# 'But no living man am I': Bioarchaeological evaluation of the first-known female burial with weapon from the 10th-century-CE Carpathian Basin

**Balázs Tihanyi**[1,2,3]*, **Kitti Maár**[3,4], **Luca Kis**[1,3], **Alexandra Gînguă**[3], **Gergely I. B. Varga**[3], **Bence Kovács**[3,4], **Oszkár Schütz**[3,4], **György Pálfi**[1], **Endre Neparáczki**[4,5], **Tibor Török**[3,4,5], **Olga Spekker**[1,5,6ʘ], **Zoltán Maróti**[3,7ʘ], **William Berthon**[1,8ʘ]

1 Department of Biological Anthropology, University of Szeged, Szeged, Hungary, 2 Department of Archaeology, University of Szeged, Szeged, Hungary, 3 Department of Archaeogenetics, Institute of Hungarian Research, Budapest, Hungary, 4 Department of Genetics, University of Szeged, Szeged, Hungary, 5 Ancient and Modern Human Genomics Competence Centre, University of Szeged, Szeged, Hungary, 6 Institute of Archaeological Sciences, Eötvös Loránd University, Budapest, Hungary, 7 Department of Pediatrics and Pediatric Health Center, University of Szeged, Szeged, Hungary, 8 Chaire d'Anthropologie Biologique Paul Broca, École Pratique des Hautes Études (EPHE), Université Paris Sciences & Lettres (PSL), Paris, France

ʘ These authors contributed equally to this work.

* balazs0421@gmail.com

**Data Availability Statement:** All relevant data are within the manuscript and its Supporting Information files.

## Abstract

Female burials equipped with weapons, a topic of interest among scholars and the general public, remain rare occurrences in archaeological records. The interpretation of such cases requires an interdisciplinary approach and a comprehensive evaluation of the available evidence, particularly regarding the sex and potential lifestyle of the deceased. Consequently, data on specific populations, regions, and time periods remain scarce. For instance, no such case has been reported before concerning the 10th century CE of the Carpathian Basin, known as the Hungarian Conquest period. Our study focuses on an interdisciplinary investigation of a previously known burial, grave No. 63 from the 10th-century-CE cemetery of Sárrétudvari–Hízóföld (eastern Hungary), which represents a unique case with grave goods including jewelry typically associated with females and archery equipment traditionally linked to males. Through archeological, anthropological, and archaeogenetic analyses, we aim to determine if this case represents the first-known female burial with weapon from the 10th-century-CE Carpathian Basin. Despite the poor bone preservation, a factor limiting data recording and evaluation, all analyses consistently indicate that the skeletal remains belonged to a female individual. The burial customs, including weapon equipment composition show analogies with male counterparts in the series. In addition, the pattern of pathological and supposed activity-related changes observed on the bones may have resulted from regular physical activity during her lifetime. In summary, our findings support the identification of this case as the first known female burial with weapon from the 10th-century-CE Carpathian Basin.

**Funding:** This work was supported by the Új Nemzeti Kiválósági Program (grant agreement no. ÚNKP-23-4-SZTE-650) of the Hungarian Ministry of Culture and Innovation (to BT) and the Cooperative Doctoral Programme for Doctoral Scholarships 2020 (grant agreement no. 1020404) of the Hungarian Ministry of Innovation and Technology (to LK). Additional support was provided by the European Research Council (ERC) under the European Union's Horizon 2020 research and innovation programme (grant agreement no. 856453 ERC-2019-SyG), the Ministry of Culture and Innovation (grant agreement MCI-670-19/2023/FÁFIN to TT), and the Competence Centre of the Life Sciences Cluster of the Centre of Excellence for Interdisciplinary Research, Development and Innovation at the University of Szeged (to TT). The funders had no role in the study design, data collection and analysis, decision to publish, or preparation of the manuscript.

**Competing interests:** The authors have declared that no competing interests exist.

## Introduction

The phenomenon of 'warrior women' has captured the attention of scholars and the wider public, leading to numerous reports and studies dating back to ancient times [1]. Although there are many legends and tales related to this topic, the number of scientifically validated archaeological cases remains limited. Interpreting such a case necessitates an interdisciplinary approach and a comprehensive evaluation of the available evidence [2], particularly regarding the sex and potential life-style of the deceased.

Biological sex is a fundamental information required for further anthropological and archaeological analyses of burials at both the individual and population levels. In traditional archaeology, the determination of sex based on the presence of artifacts previously associated with males or females remains a widely used and accepted approach. However, this is a consequence of the loss of distinction between the terms "sex" and "gender" often experienced in the anthropological, archaeological, and medical literature (e.g., [3, 4]).

Sex refers to a person's biological identity (e.g., [3, 4]). While both macroscopic and molecular techniques are available for investigation, traditional anthropological methods are more prevalent due to their relative speed, and cost-effectiveness compared to molecular techniques. In biological anthropology, the estimation of sex relies on human skeletal dimorphism [4]. Numerous metric and morphological methods have been developed to analyze sexual differences in various anatomical regions, such as the pelvic girdle, skull, and long bones (see e.g., [4–23]). Among these methods, the most reliable ones (with probabilities reaching up to 95%) focus on the pelvic girdle (e.g., [11–13, 23]). However, despite their high level of accuracy, several internal and external factors can influence the results of sex determination. The main limitation lies in the qualitative and quantitative state of preservation of the bones. Poor preservation of skeletal remains often leads to indeterminate or unknown sex, and depending on the researcher's approach, it can even result in misclassification and overinterpretation of data. Additionally, determining the sex of sub-adult individuals using anthropological methods is still a controversial issue, as the secondary sexual differences in their skeletons are not yet fully developed (e.g., [19, 24]). In recent years, biomolecular approaches to sex determination have also become common in bioarchaeology [25, 26]. While techniques focusing on the X-linked amelogenin gene and the pseudo amelogenin gene on the Y-chromosome have been used for decades, their authenticity was criticized by many scholars (see e.g., [27]). The most reliable methods are based on the paleoproteomic analysis of sexual dimorphic proteins (e.g., [26, 28, 29]) and genomic techniques that calculate read ratios mapping to X and Y chromosomes or rely on the ratios of X chromosome-derived data to the autosomal coverage (e.g., [27, 30, 31]). However, molecular techniques also have limitations, mainly related to fragmentation and modern human contamination (e.g., [26, 27, 30]). Nevertheless, they have been successfully applied, even in cases where morphological methods were insufficient (e.g., [26, 32]).

On the other hand, gender, as the cultural interpretation of sexual difference, is an aspect of a person's social identity (e.g., [3, 4, 33, 34]). It is a determining factor that influences various aspects of life (e.g., social organization) and death, specifically mortuary practices of present and past societies, providing essential tools for archaeological research [35]. Although invaluable contributions have been made in gender-related archaeology in the past decades (see, e.g., [2, 33, 36–40]), gender still cannot be considered an integrated aspect in studies evaluating past societies [40]. This is unfortunate since these studies have revealed that investigating sex- and gender-related questions is a complex problem. Furthermore, scholars have already emphasized that determining sex based solely on the presence of artifacts previously associated with males or females may be insufficient (e.g., [34, 41]). Even in the case of weapons, which are traditionally considered as male attributes, researchers must be cautious with the

interpretation, as female burials with weapon-related grave goods are also documented from the Paleolithic period to modern ages (see e.g., [1, 2, 42–48]). Although the number of such cases and related studies has increased significantly, data on specific populations, regions, and time periods remain scarce. Consequently, female burials with weapons, in general, are still considered to be rare or even unique phenomena [1, 43].

Burials with weapon deposits–regardless of the biological sex of the deceased–are often interpreted as warrior graves. However, scholars have revealed certain methodological problems with this approach (e.g., [49–52]). Warriors belonged to a social class defined by specific characteristics on social and legal levels. Many of these social and legal characteristics are invisible or obscured by other factors (e.g., economic or religious) in the archaeological material. Consequently, scholars have emphasized that archaeological methods alone are insufficient for determining many aspects of social and legal groups (e.g., [53–56]). Additionally, the simple evaluation of grave goods as indicators of lifestyle and 'mirrors of life' has also been debated (e.g., [57, 58]). In some cases, the complex bioarchaeological analysis has revealed that the presence or absence of weapon equipment does not straightforwardly indicate the possible military role of the deceased (e.g., [52, 57–59]). On the other hand, warriors, defined as individuals practicing specific physical activities, may suffer or develop certain musculo-skeletal changes that can be analyzed with bioanthropological methods. Traumatic injuries such as blunt force or sharp force traumas are commonly associated with warfare and warriors as participants in interpersonal conflicts and violence. However, investigating violence, warfare, and warriors is a more complex bioarchaeological issue (e.g., [52, 60, 61]). These traumas specifically relate to interpersonal violence, but it is possible that "civilian" individuals, other than dedicated warriors, could also have suffered from such injuries [2, 61]. In addition to evaluating acute traumas, another anthropological approach involves the investigation of skeletal markers associated with repetitive physical stress, taking into account that bones can adapt their structure and form depending on the mechanical loading [62]. Activity-related skeletal changes include quantitative and qualitative features, with entheseal changes (at the insertion sites of tendons and ligaments), joint changes, and bone geometry (i.e., alterations in the shape and robusticity of bones) being the most frequently studied parameters [63–66]. Numerous studies have focused on the analysis of changes related to general or specific activities (e.g., [59, 67–83]). Additionally, the investigation of hunting- and warfare-related activities has yielded promising results, as regular practice with specific types of weapons (e.g., melee weapons, atlatl, and bow) can result in the development of various skeletal changes (e.g., [67, 83–87]). In ideal conditions, differences between individuals buried with and without weapons can be detected at the population level [72]. However, the link between the actual activity and the skeletal changes is not yet clear, as non-mechanical factors (e.g., genetics, sex, age, and metabolic disorders) can also influence the development of these bony changes (e.g., [72, 88–93]). Although physical stress is still considered one of the main factors influencing the development of these changes [94, 95], researchers must control for possible influences of non-mechanical factors and avoid overinterpretations [72, 78].

Regarding papers describing 'female warrior' burials, some of the results have been excessively criticized (see e.g., [45] supplementary material), indicating that gender-related investigations are not yet mainstream [40]. However, a proper evaluation of this topic requires, indeed, several archaeological and biological considerations [2]. It should be acknowledged that while archaeological and historical analyses play a central role in related studies, bioarchaeological examination of the skeletal remains (e.g., age estimation, sex determination, and evaluation of pathological and activity-related changes) has been conducted to a lesser extent [1]. Consequently, in those studies with limited or no bioarchaeological data, the main findings, such as 'female' and 'warrior' are often overinterpreted and lack methodological support.

In Hungarian archaeology, research on gender has been considered rudimentary, but the number of studies related to gender has increased since the 2000s (e.g., [96–104]). Additionally, female burials with weapons have been reported from various time periods and regions of the Carpathian Basin, such as the Sarmatian period (ca. 1st–5th centuries CE) and the Gepid period (ca. second half of the 5th century–567 CE) of the Great Hungarian Plain [105, 106], the Langobard period (ca. 510–568 CE) of Transdanubia (e.g., [105]) or the Avar period (ca. 567–9th century CE) of the Carpathian Basin (e.g., [107, 108]). However, these findings mostly consist of fragmented artifacts, individual pieces (e.g., arrowhead) or armor fragments (e.g., chain-mail fragments), which differ from the typical weapon equipment found in the cemeteries of these populations. Due to the complexity of their interpretation, these burials have not been classified as warrior graves, and alternative hypotheses explaining the presence of these objects in the burials (e.g., amulets) have not been ruled out (e.g., [105, 106]). This is also the case for the 10th-century-CE archaeological horizon known as the Hungarian Conquest period in the Carpathian Basin. The Hungarians, or Magyars, originating from the steppe region, arrived in the Lower Danube region in the 830s CE and moved to the Carpathian Basin at the end of the 9th and beginning of the 10th centuries CE (e.g., [109–111]). They quickly established themselves in the new territories and integrated local populations such as the Avars and Slavs (e.g., [110, 112]), eventually leading to the formation of the Christian Hungarian Kingdom at the turn of the 10th and 11th centuries CE (for a brief summary, see Methods S1 in [113]). During the 9th–10th centuries CE, the feared Hungarian mounted archers were involved in numerous battles and conducted their own attacking campaigns throughout Europe (e.g., [114]), which might be reflected in the cemeteries, as burials with weapons are characteristic of the 10th-century-CE archaeological horizon in the Carpathian Basin (e.g., [115, 116]). Weapon deposits in these graves consist of melee weapons such as axes, spears, sabers, swords, and swords with saber hilts (e.g., [116–126]), as well as archery equipment, particularly arrows (preserved arrowheads), composite bows (preserved antler plates), quivers, and bow cases (preserved iron parts, and metal or antler ornaments) (e.g., [116, 118, 127–140]). However, the appearance and distribution of these weapon types in the graves vary, and archery equipment has been found in significantly higher frequencies compared to melee weapons (e.g., [115, 141]). The investigation of these artifacts and burials is hindered by two factors: a) in most cases, only the weapon parts made of inorganic materials have been preserved due to the soil and climate conditions of the Carpathian Basin and b) according to our current knowledge, no correlation has been found between the possible military rank of the deceased, and the number and type of weapons placed in the grave [119].

Weapon equipment is predominantly found in adult burials. Although, single arrowheads have been discovered in several female graves, these have never been interpreted as weapons but rather as amulets (e.g., [103, 142, 143]). A 10th-century-CE female burial with artifacts considered as weapons, such as melee weapons or complete archery equipment (bow, quiver, and arrow) has never been previously described in the Carpathian Basin.

As part of larger anthropological (e.g., [59, 75, 144, 145]) and archaeogenetic (e.g., [113, 146]) research projects, we conducted a bioarchaeological re-evaluation of the Sárrétudvari–Hízóföld series (Hajdú-Bihar county, Hungary), which is the largest known 10th-century-CE cemetery in the Carpathian Basin [147]. In this paper, we focus on a specific burial, grave No. 63, which contained a weapon deposit, and our earlier data suggested that the deceased was a female (Supplementary material in [146]). We conducted an extensive interdisciplinary analysis using archaeological, anthropological, and archaeogenetic methods to determine if this burial is indeed the first-known female burial with weapon from the 10th-century-CE Carpathian Basin.

## Material and methods

### The cemetery of Sárrétudvari–Hízóföld and the description of burial No. 63

Due to intensive agricultural activity near the present-day village of Sárrétudvari (Hajdú-Bihar county, Hungary) (Fig 1A), archaeological excavations were carried out at the Sárrétudvari–Hízóföld site (SH) between 1983 and 1985. These rescue excavations revealed several Bronze Age burials and a large cemetery containing 262 graves from the Hungarian Conquest period [148, 149]. Since the initial archaeological [148, 149] and anthropological (e.g., [150, 151]) analyses, the 10th-century-CE cemetery of Sárrétudvari–Hízóföld has undergone extensive archaeological (e.g., [147, 152–154]) and bioarchaeological, particularly anthropological (e.g., [59, 75, 78, 144, 145, 155–158]) and archaeogenetic (e.g., [113, 146, 159–163]) investigations.

The 262 excavated graves contained the skeletal remains of 263 individuals (in addition to bone fragments of two individuals that were too damaged for examination). Among these individuals, 101 were sub-adults and 162 were adults. Initially, 70 females and 85 males were identified using anthropological methods that focused on both the skull and postcranial skeleton [155]. Later, a reassessment of the age at death and sex of the SH individuals confirmed the biological sex of 52 females and 69 males through additional anthropological methods focusing only on the pelvis [144] and archaeogenetic analysis of a representative sample [113, 146].

The graves were generally oriented in the west-east direction with minor variations to north-south and in a lower extent to south-north directions. Some of the graves showed signs of contemporary disturbance. Although most skeletons were found in an extended position, lying on their backs, two cases with bent knees have also been documented [149].

The archaeological findings included various types of jewelry (e.g., penannular hair rings, earrings, strings of beads, bracelets, and finger rings), clothing elements (e.g., belt buckles and bell buttons) and dress fittings (metal ornaments), tools (e.g., knives and fire-lighting tools), horse riding-related equipment (stirrups, bits, and saddle parts), horse bones, and weapons [149]. Notably, the number of burials with weapons discovered– 58 in total–surpassed that of other cemeteries from this period in the Carpathian Basin. These weapons included archery equipment (arrowheads, traces of quiver, and antler bow plates), and in three cases, an additional saber or axe [149].

Among these armed burials, grave No. 63, located at the western border of the cemetery, was oriented in the southwest–northeast direction (230.6˚–50.6˚) and fortunately remained undisturbed by contemporary or modern activities [149]. However, traces of minor *post mortem* disturbance were detected (possibly by animals) as some bones were not in anatomical position (e.g., left clavicle, left ulna, and right radius). The single burial contained a skeleton partly lying on its right side with slightly bent knees (Fig 2B). The grave goods consisted of:

a) A silver penannular hair ring near the left part of the occipital bone (Fig 1B/3.);

b) Three bell buttons–one was positioned beneath the skull, another beneath the right clavicle, and the third close to the knees (Fig 1B/2, Fig 1B/5);

c) A string of beads near the left clavicle, including faience beads with blue eye-shaped inlays, yellow and white semiprecious stone beads, and segmented glass beads in various colors (Fig 1B/4);

d) An "armor-piercing" arrowhead found at the distal end of the grave pit (Fig 1B/1). Iron fragments possibly belonging to further arrowheads were also found in the soil of the grave;

e) Fragmented iron parts of a quiver situated near the left side of the skeleton from the shoulder to the toes; and

f) An antler bow plate with convex sides and peaked ends (Fig 1B/6) located near the hip and the left hand, appearing as if "the deceased was gripping the bow" [149].

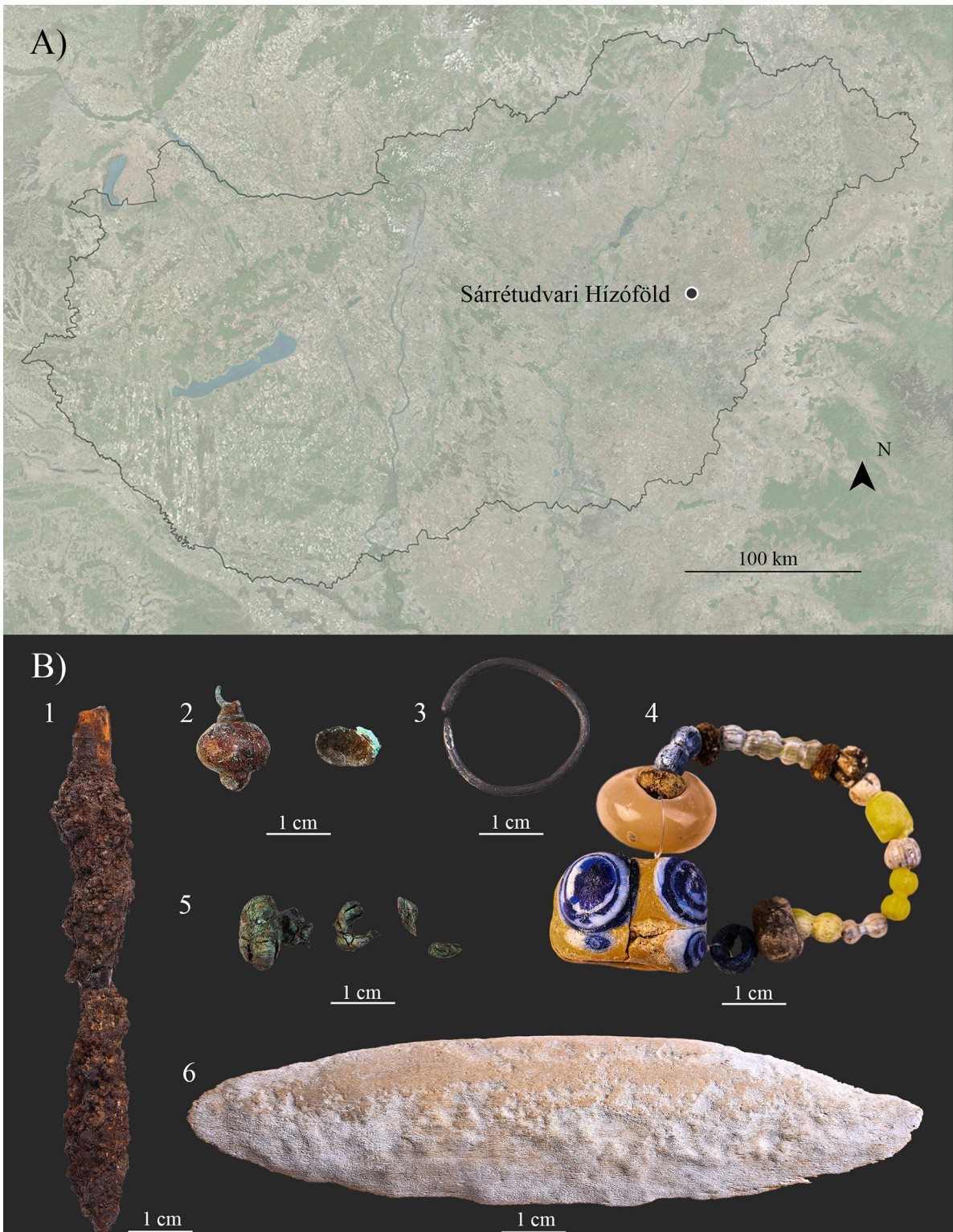

**Fig 1.** A) Map of Hungary showing the location of the Sárrétudvari–Hízóföld archaeological site (created in QGIS (version 3.16.16; https://www.qgis.org/), and edited by Luca Kis). B) Artifacts found in grave No. 63: 1) arrowhead; 2) bell button; 3) silver penannular hair ring; 4) a string of beads; 5) fragments of bell buttons; and 6) antler bow plate (photos taken by Zoltán Faur and edited by Luca Kis).

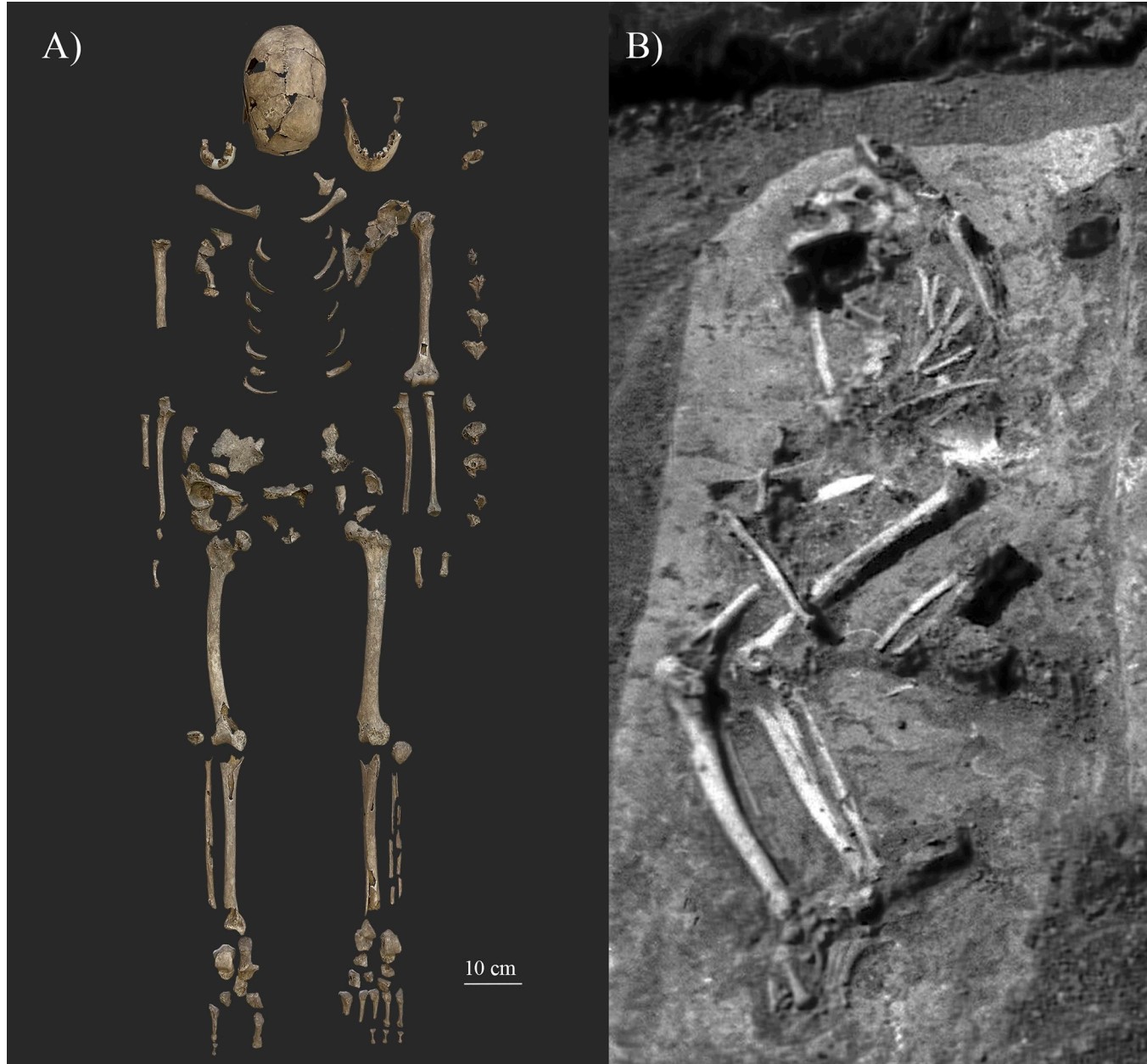

**Fig 2. The skeletal remains discovered in grave No. 63.** A) Photo indicating the current state of preservation of the bones (photo by Luca Kis); and B) Photo of the burial No. 63 *in situ* (photo taken by Ibolya M Nepper, edited by Luca Kis).

The skeletal remains of the individual (SH-63) recovered from the grave are in poor condition (Fig 2A). According to the criteria described by Spekker et al. [164], the cranium is relatively complete (50–80% is extant) and fairly preserved, with moderate damage to the surfaces. However, the facial skeleton is highly incomplete and poorly preserved, with only the mandible, fragments of the two maxillae, and the left zygomatic bone present. The postcranial skeleton is relatively complete (50–80% of the bones are extant) but poorly preserved, especially the bones of the spine (e.g., vertebral bodies), thoracic wall (e.g., sternum), and pelvis (e.g., sacrum and ilium), which are mostly represented by of bone fragments. SH-63 can be identified as an

adult individual since all extant bones were completely fused, including the vertebral bodies, sternal end of the clavicle, and fragments of the iliac crest of the coxal bone. Specific markers, such as the obliteration of the endocranial sutures ([165]) and general characteristics of the bones (e.g., thinness of the cranial bones) [4, 19, 166], suggest that the individual belonged to the middle-aged or old adult category. However, providing a more precise age-at-death is unachievable due to the severe *post mortem* damage or complete absence of most bone parts used for age estimation (e.g., auricular surface or symphysial surface).

## Anthropological re-examination of the skeletal remains

As part of our comprehensive investigation, we conducted an anthropological analysis on SH-63, focusing specifically on sex determination and evaluation of activity-related skeletal changes. Although multiple metric and macromorphological methods for sex determination of adult skeletal remains (e.g., [5, 9, 11–13, 15, 19, 20, 23]) were considered, only methods examining the morphological traits of the skull were applicable, as the analysis of the coxal bones provided insufficient data for metric or morphological evaluations. Data were recorded based on the standards provided in [5, 19], and results were discussed following the descriptions of [5, 8, 9, 19]. In addition, macromorphological methods were used to assess the supposed activity-related skeletal changes of the musculoskeletal system, particularly joint changes [90], entheseal changes [68], morphological variants [4, 167], and traumas [166, 168]. We also followed standard macromorphology-based osteological and paleopathological research methods and categorization systems (e.g., [90, 166, 169–171]) to record and identify any pathological changes present on the bones that might have influenced the individual's lifestyle and the development of activity-related skeletal changes. Generally, a rule was applied whereby at least 50% of the studied bone surface or part needed to be present for scoring (e.g., [70, 82]). However, our primary focus was on describing the changes observed on the extant bones rather than scoring them on an ordinal scale, given the poor preservation of the bones and the lack of comparative data on females with or without weapons from this era. Therefore, changes that met the criteria outlined in the cited studies were noted as 'presence of changes' and were described in detail, even if less than 50% of the relevant area was present.

## Archaeogenetic examination of the skeletal remains

The pre-PCR procedures were conducted in the specialized ancient DNA facilities of the Department of Genetics, University of Szeged and the Department of Archaeogenetics, Institute of Hungarian Research, Hungary. Mitogenome data of the sample utilized in this study had already been published ([146]) and now we performed shallow shotgun sequencing. Samples were obtained from SH-63 for molecular analysis, including a multirooted tooth from the mandible (referred to as the tooth sample), a sample from the petrous part of the right temporal bone (referred to as the petrosa sample), and a piece from the left humerus (referred to as the humerus sample). A detailed description of the genetic analysis can be found in S1 Text.

Biological sex was first assessed with the method described by Skoglund and colleagues [27]. In addition, we used the Rx method first published by Mittnik and colleagues and modified later by de Flamingh and colleagues, which can give accurate results from several thousands of reads mapping to the human genome [30, 31].

## Ethics statement

Specimen number: SH-63 (inventory no. 10793; grave no. 63). The skeleton evaluated in the described study is housed in the Department of Biological Anthropology, University of Szeged,

in Szeged, Hungary. Access to the specimen was granted by the Department of Biological Anthropology, University of Szeged (Közép fasor 52, H-6726 Szeged, Hungary).

No permits were required for the described study, which complied with all relevant regulations. The research has been conducted in an ethically responsible manner–the bone remains of SH-63 have been examined with dignity and respect.

## Results

### Sex determination of SH-63

Generally, the poor preservation of the bones limited the anthropological sex determination. Only methods examining the skull were applicable, as the analysis of the coxal bones provided insufficient data for metric or morphological evaluations. Although the mandible showed slight eversion and rugosity of the gonial angle, the cranial features widely used for sex determination showed feminine characteristics. For instance, a very small mastoid process (score -2 in [5] and score 1 in [19]) (Fig 3A), a smooth frontal contour with little or no projection of the glabellar area (score -2 in [5] and score 1 in [19]) (Fig 3B), the complete absence of the nuchal crest with a smooth external occipital surface (score -1 in [5] and score 2 in [19]) (Fig 3C), and

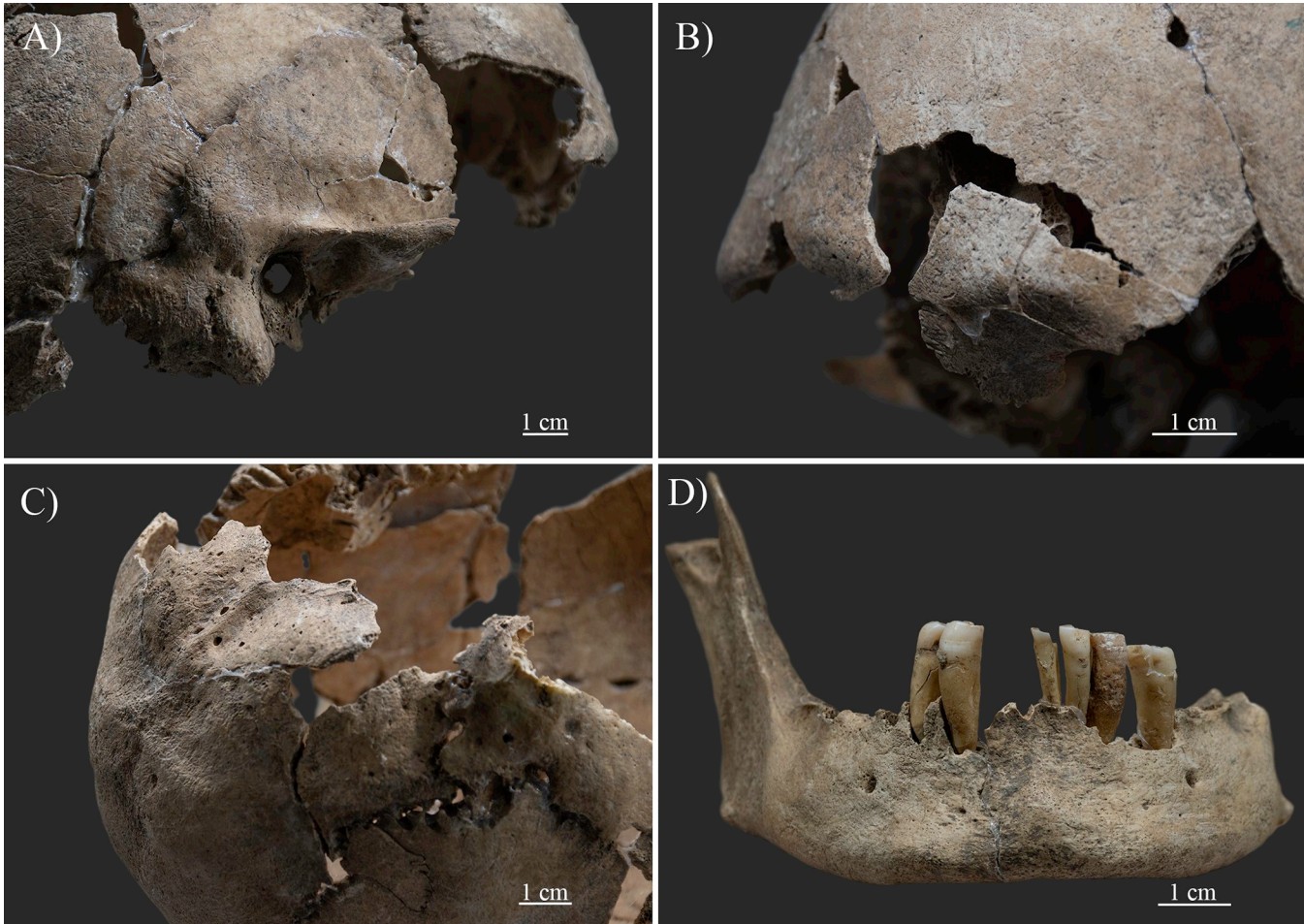

**Fig 3. Cranial features used for sex determination showing feminine characteristics (photos by Luca Kis).** A) A very small mastoid process; B) A smooth frontal contour with little or no projection of the glabellar area; C) Complete absence of nuchal crest with a smooth external occipital surface; and D) A smooth mental eminence with slight projection above the surrounding bone.

**Table 1. Shotgun sequencing statistics of SH-63.**

| Sample (ID) | All reads | Endogen reads | Endogen percent | Unique endo reads | Duplication percent | Library size | Insert size | Insert standard deviation |
|---|---|---|---|---|---|---|---|---|
| SH-63 pars petrosa (SH63P) | 118334 | 95626 | 0.808 | 92512 | 0.033 | 1444142 | 61.71 | 25.54 |
| SH-63 tooth (SH63F) | 176237 | 1654 | 0.009 | 1608 | 0.028 | 25399 | 82.39 | 41.17 |
| SH-63 humerus (SH63V) | 125486 | 482 | 0.004 | 466 | 0.033 | 5584 | 59.01 | 25.48 |

a smooth mental eminence with slight projection above the surrounding bone (score -1 or 0 in [5] and score 2 or 3 in [19]) (Fig 3D) were detected.

The results of the genetic analysis are summarized in Tables 1–3. The poor state of preservation hindered the genetic analysis, especially in the tooth and humerus samples, which contained a very low percentage of endogenous DNA (0.9% and 0.4%, respectively) (Table 1). Nevertheless, the petrosa sample had a sufficiently high percentage of endogenous DNA (80.8%), with an average fragment length of ca. 60–80 bp. The sex determination of tooth and humerus samples using the method by Skoglund and colleagues resulted in an assignment consistent with female (XX) due to the very low number of reads. Meanwhile, analysis of the petrosa sample determined the individual as female (XX) with full confidence (Table 2). Additionally, all three samples were assigned as female by the Rx method, a method that is validated to work from even a few thousand reads (Table 3).

## Pathological and activity-related skeletal changes

A general feature observed in the skeleton of SH-63 is the relatively lightweight nature of the bones. Additionally, other general conditions were noted in the skeletal remains, such as bone fragility, traces of a reduced trabecular system in the vertebrae, an increased diameter of the medullary cavity in the long bones (Fig 4B), and thinning of cortical bone in both the skull and postcranial elements [90, 166]. For instance, the thickness of the parietal bones at the top of the calvaria, measured with a spreading caliper, did not exceed 3 mm (Fig 4A), and the cortical bone thickness of the right humerus, measured approximately at the area of the midshaft, was between 1–2 mm (Fig 4B).

A well-defined oblique line is present at the proximal end of the right humerus, at the level of the surgical neck (Fig 5). This line is marked with sharp exostoses (0.1–0.2 cm in size), porous new bone formations, pitting, and at least three relatively large (ca. 0.6x0.7 cm, 0.8x0.3 cm, and 1.1x0.2 cm in size with an average depth of 0.1–0.2 cm) impression-like osteolytic lesions with macroporosity and pitting on their surface, as well as a slight medial and posterior angulation of the proximal bone end. The remodeling of the surface is more pronounced at the anterior and medial parts of the humeral neck, while the posterior and lateral parts are relatively smooth, with only remnants of a sharp margin detectable on the surface. The observable insertion sites of the main muscles of the arm and trunk (e.g., deltoid, pectoralis major, latissimus dorsi, and teres major muscles) were not affected by the related changes.

**Table 2. Results of the genetic sex determination with the method described by Skoglund and colleagues ([27]).**

| Sample (ID) | Nseqs | NchrY+NchrX | NchrY | R_y | SE | 95% CI | Assignment |
|---|---|---|---|---|---|---|---|
| SH-63 pars petrosa (SH63P) | 3948 | 3948 | 8 | 0.002 | 0.0007 | 0.0006–0.0034 | XX |
| SH-63 tooth (SH63F) | 72 | 72 | 0 | 0.0 | 0.0 | 0.0–0.0 | consistent with XX |
| SH-63 humerus (SH63V) | 20 | 20 | 0 | 0.0 | 0.0 | 0.0–0.0 | consistent with XX |

**Table 3. Results of the genetic sex determination with the Rx method described by Mittnik and colleagues ([30]) and modified by de Flamingh and co-workers ([31]).**

| Sample (ID) | Nseq | NchrX | NchrY | p-value | Rx | 95% CI | Assignment |
|---|---|---|---|---|---|---|---|
| SH-63 pars petrosa (SH63P) | 3948 | 3940 | 8 | 1.001e-15 | 0.9977402 | 0.9508828–1.044598 | XX |
| SH-63 tooth (SH63F) | 72 | 72 | 0 | 1.029e-12 | 1.095995 | 1.028932–1.163057 | XX |
| SH-63 postcranial (SH63V) | 20 | 20 | 0 | 1.515e-06 | 1.360008 | 1.051178–1.668838 | XX |

Additionally, an oblique fracture line is also observed on the lateral margin of the right scapula (ca. 3.5 cm inferior to the infraglenoid tubercle), which is consistent with the described lesions of the humerus (Fig 6). Two separate bone fragments are present due to the complete discontinuity of the lateral margin (Fig 6A). The margins of the bone segments along the two sides of the fracture line are rounded and thickened, forming a secondary facet (ca. 1.7x0.7 cm) with a porotic surface between them (Fig 6B). However, most of the scapular body is missing, and the extent and further characteristics of the lesion are unknown.

Similar bony changes are also present at the lateral margin of the left scapula (Fig 7). Discontinuity, namely a complete fracture, is observable around the middle part of the lateral margin (ca. 3.5 cm inferior to the infraglenoid tubercle), dividing the bone into at least three distinctive parts (Fig 7A). The inferior part of the lateral margin was dislocated and partly moved behind the superior part, into a postero-superior direction with a possible posterior and medial angulation compared to its original position. Additionally, a relatively large (4.2x2.6 cm in size) separate segment of the scapular body (from the subscapular/infraspinatus fossa) is also preserved with a deformed shape and contour, wedged between the two fragments of the lateral margin (Fig 7A & 7C). A porous, thick new bone formation (with a ca. 1–1.5 cm antero-posterior width) is present between the bone segments, which probably partially fused the bone parts in this secondary position (Fig 7B & 7C). The margins of the bone parts around the fracture are rounded and covered with porous new bone layers. In addition, insertion sites of the subscapularis muscle at the costal surface and the infraspinatus and teres minor muscles at the dorsal surface were also affected by the deformation and fixation of the fractured bone parts in a secondary position. Nevertheless, the insertion sites of the main

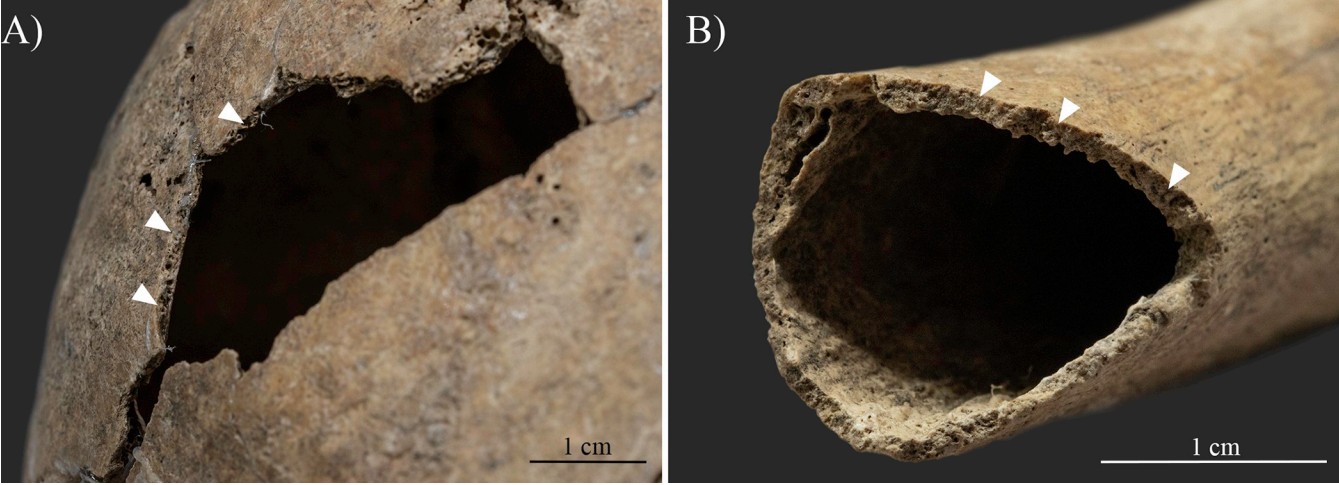

**Fig 4. General conditions observed in the skeletal remains (photos by Luca Kis).** A) Thinning of cortical bone in the skull; and B) Thinning of cortical bone and increased diameter of the medullary cavity in the right humerus.

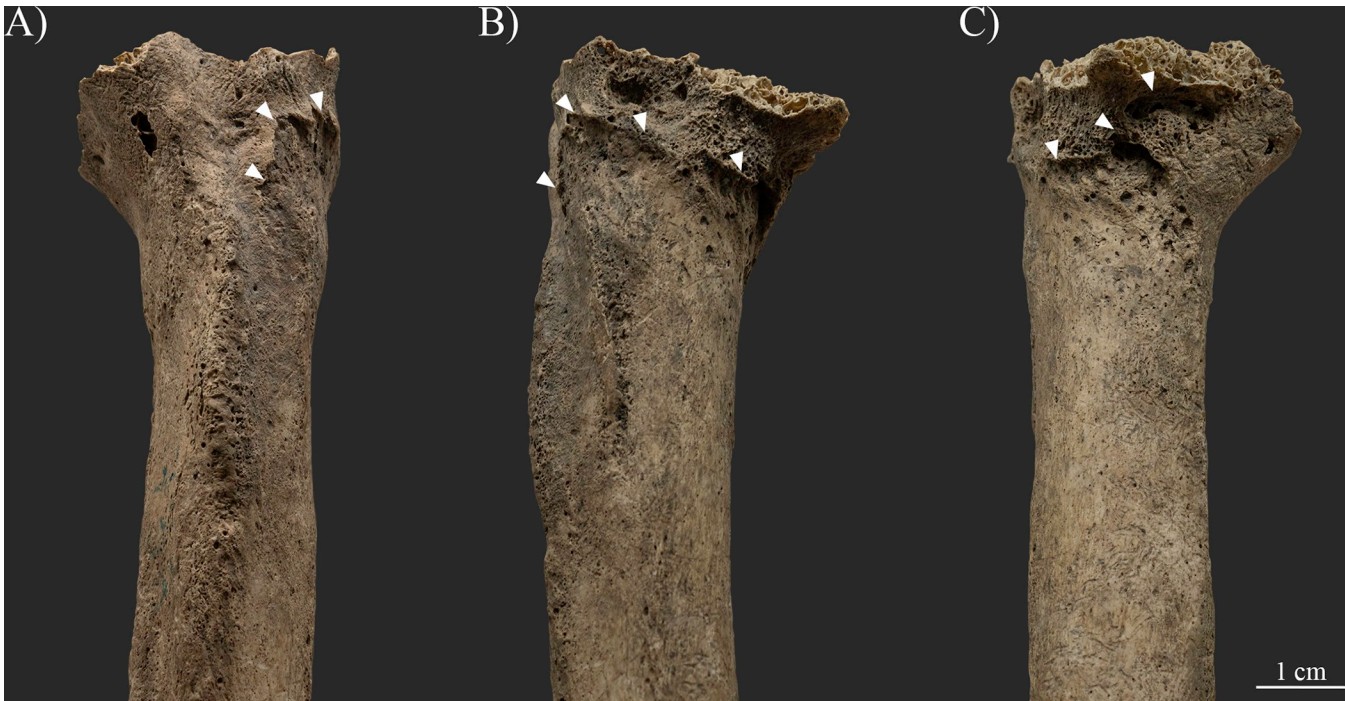

**Fig 5. Changes observed on the proximal end of the right humerus, at the level of the surgical neck (photos by Luca Kis).** A) Lateral and anterior views; B) Anterior and medial views; and C) Medial and posterior views.

muscle groups of the trunk, shoulder, and arm (e.g., deltoid, latissimus dorsi, teres major, triceps brachii, and biceps brachii muscles) were not affected. Additionally, neither the left humerus, nor the available rib fragments were affected by similar changes.

Further changes in the musculoskeletal system, particularly joint changes and entheseal changes, were observed in the skeleton (S1 and S2 Tables). While joint changes were widely observed on the extant bone elements, these were predominantly present in the bones of the upper limbs, affecting both the elbow and shoulder (S1 Table). The proximal part of the upper extremities was more affected, with osteophytes present on more than 50% of the scapular margin, and complete remodeling of the articular surface on the sternal end of the right clavicle was detected (Fig 8). In addition, remodeling of the postero-superior acetabular rim was noted on both coxal bones.

We identified entheseal changes on both the upper and lower limb bones, at the insertion sites of the main muscle groups of the arms, trunk, hip, and legs (Fig 9 and S2 Table). Characteristic changes are present on the linea aspera of the femur (insertion site of the vastus medialis, vastus lateralis, adductor longus, adductor brevis, adductor magnus, and biceps femoris muscles) (Fig 9D), the rugosity for the deltoid muscle on the clavicle (Fig 9C), the ulnar tuberosity (insertion site of brachialis muscle) (Fig 9B), as well as the supinator crest of the ulna and the lateral surface of the proximal third of the radius (at the level between the radial tuberosity and interosseus border), which are the insertion sites of the supinator muscle (Fig 9A). Furthermore, alterations in the general morphology, specifically medial elevation of the radial tuberosity, were observed, resulting in an ear- or lip-shaped formation of the enthesis.

The changes appeared mostly bilaterally where the insertion sites were preserved on both the left and right sides. However, slight differences were noted (Fig 10). For instance, in the case of the crest of the lesser tubercle of the humerus (insertion site of the latissimus dorsi/

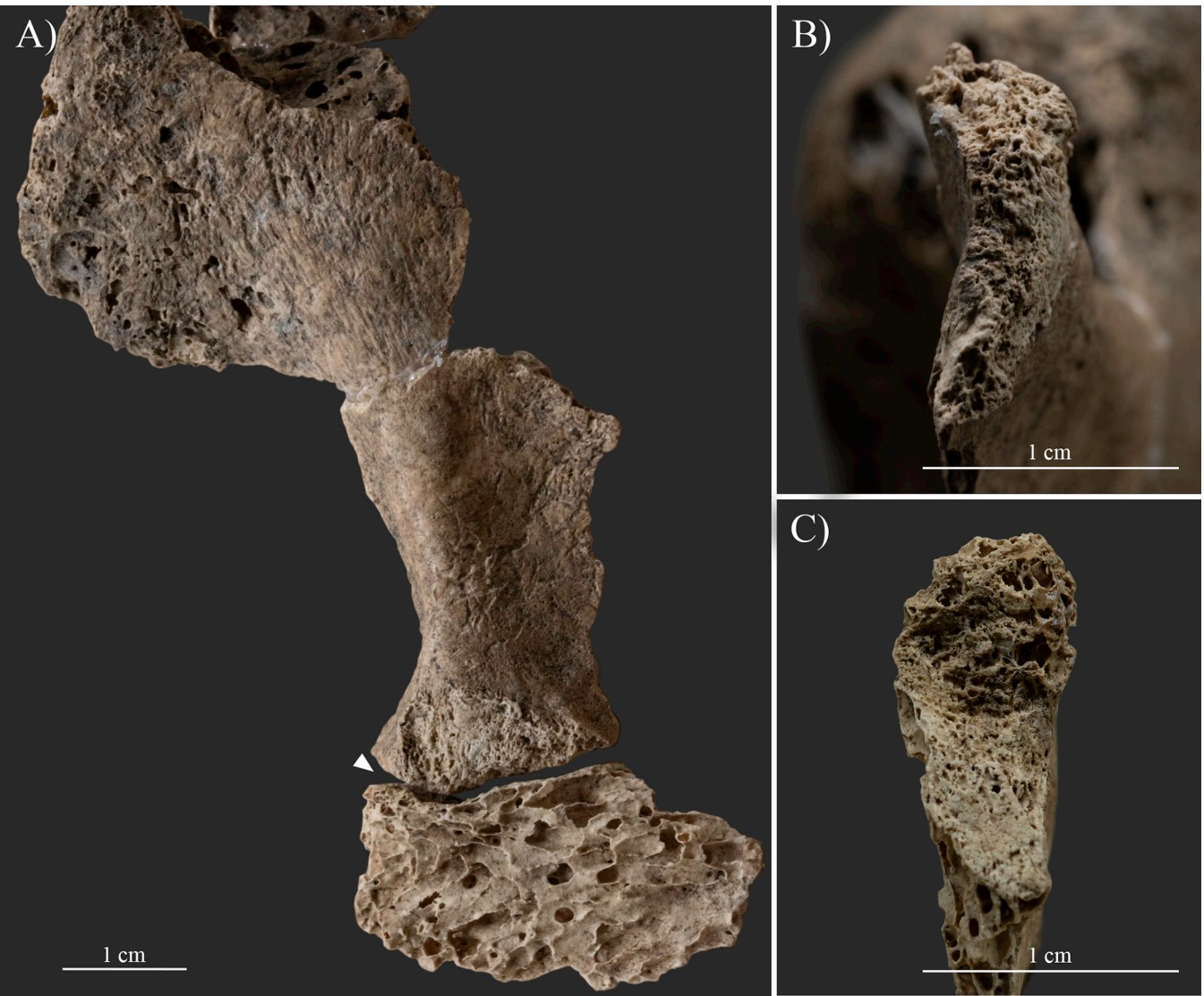

**Fig 6. Changes observed on the lateral margin of the right scapula (photos by Luca Kis).** A) The two separate bone fragments of the lateral margin (anterior view). Note: the glenoid cavity is in the upper left corner of the photo; and B) The margins of the two bone fragments forming a secondary facet.

teres major muscles) (Fig 10C), the radial tuberosity of the radius (insertion site of the biceps brachii muscle) (Fig 10A), and olecranon of the ulna (insertion site of the triceps brachii muscle) (Fig 10B), the right side showed more pronounced marginal and/or surface changes compared to the left side.

In addition, an imprint with a slightly rough surface and a bony rim were observed on the anterior margin of the left femoral neck (the right was not observable), which was slightly lower than the frontal plane of the femoral neck (Fig 11).

## Discussion and conclusions

### Bioarchaeological evaluation of SH-63

In our investigation, the primary objective was to determine the biological sex of SH-63 using both anthropological and archaeogenetic methods. It is important to note that the

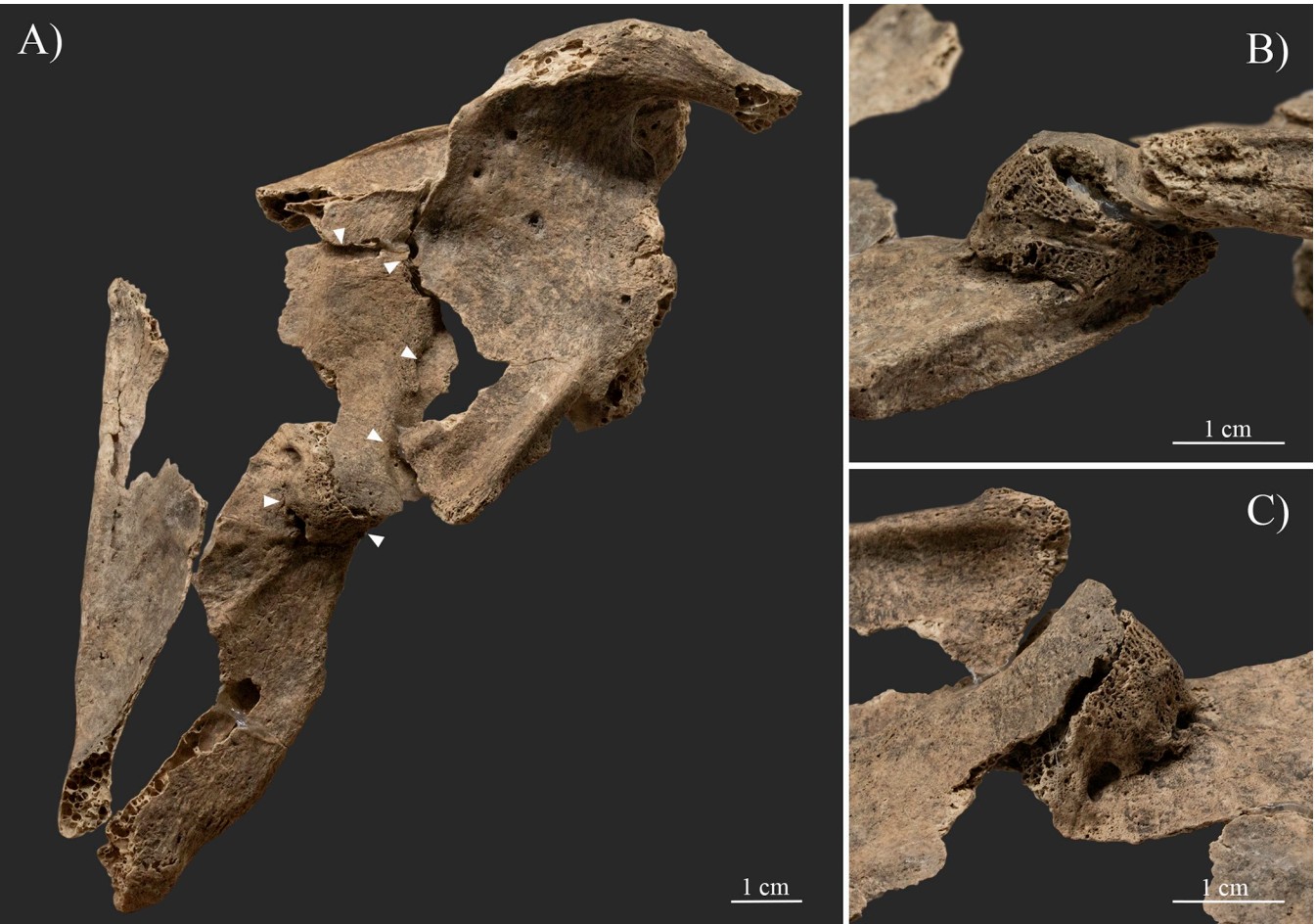

**Fig 7. Changes observed on the left scapula (photos by Luca Kis).** A) The complete fracture (white arrows) present on the middle section of the scapula, dividing the bone into at least three distinctive parts; B) The porous, thick new bone formation present between the bone segments at the middle section of the lateral margin; and C) Close-up of the separate bone segment wedged between the two fragments of the lateral margin and the porous, thick new bone formation present between the bone segments.

anthropological analysis lacks full confidence due to the poor state of preservation of the bones, especially the absence of data concerning the pelvic region. Nevertheless, most sexually dimorphic traits of the skull exhibited feminine characteristics. The application of sex determination methods that evaluate the skull ([5], [8], [9], [19]) suggests that SH-63 can be classified as a probable female individual, following the terminology used by [4]. This finding appears to contradict earlier results of anthropological studies (see e.g., [149], [151]).

Despite the poor state of preservation of the bones, which limited the archaeogenetic analysis of SH-63, the percentage of endogenous DNA and read numbers of the petrosa sample (80.8% and 95626 respectively) were sufficiently high for an unambiguous assessment. The tooth (0.9%) and humerus samples (0.4%) contained minimal amounts of endogenous DNA. While the analysis of these two samples on their own lacks confidence, the assignment of the samples is consistent with a female, supporting the results derived from the analysis of the petrosa sample. Additionally, the average fragment length and the empirical PMD pattern (S1 File) of endogenous DNA is consistent with the characteristics of fragmented archaic DNA samples (e.g., [172]). Accordingly, contamination from modern human DNA is unlikely to bias sex determination. The individual was confidently assigned as female using both the Rx

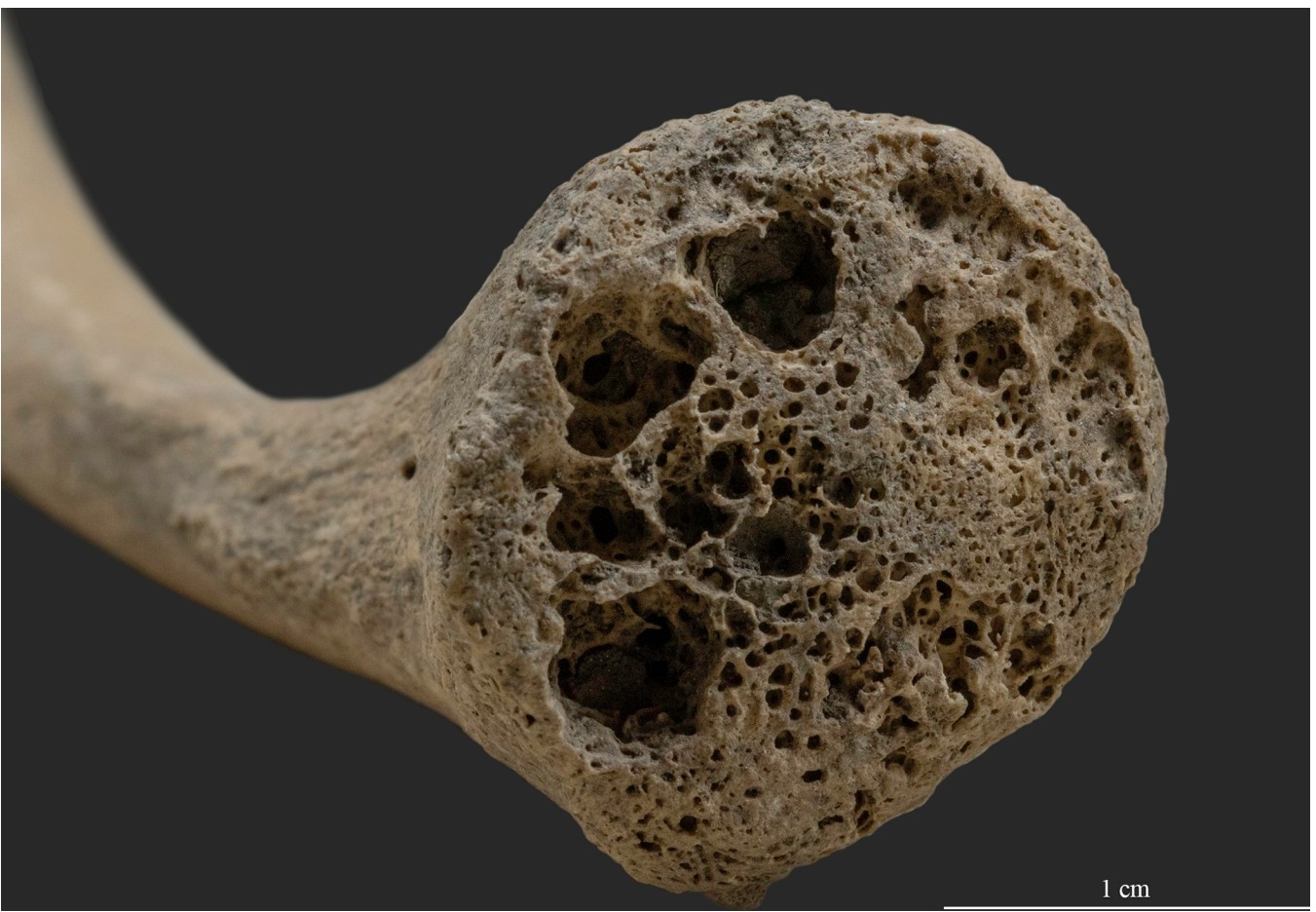

**Fig 8. Joint changes observed on the sternal end of the right clavicle (photo by Luca Kis).**

method and the method described by Skoglund and colleagues [27, 30]. Thus, the archaeogenetic sex determination of the three different samples are consistent with each other, and the results are supported by two independent and robust statistical tests widely used in genetic investigations of archaic samples (e.g., [25, 173]). Therefore, the re-examination of SH-63 confirmed the earlier genetic findings [146]. Ultimately, all observations and analyses consistently suggest that the skeletal remains belonged to a female.

Several antemortem features are present on the skeletal remains of SH-63. Firstly, we observed the lightweight nature of the bones, which could indicate osteopenia, a loss of bone mass. Specifically, bone fragility, traces of a reduced trabecular system in the vertebrae, an increased diameter of the medullary cavity in the long bones, thinning of cortical bone in both the skull and postcranial elements, and antemortem bone fractures may be associated with the presence of osteoporosis in the skeleton (e.g., [166, 174–177]). The presence of osteoporosis, described as a disease that typically affects older women [166, 175], could indirectly support the results concerning the sex determination of SH-63. However, the assessment of the registered features is limited for two main reasons. Firstly, diagnosing osteoporosis requires consideration of several methodological factors, including the use of quantitative methods (e.g., radiometry or histomorphometry) [178, 179]. As no radiological or histological investigations could be conducted during the analysis of SH-63, there is no such evidence for a definitive

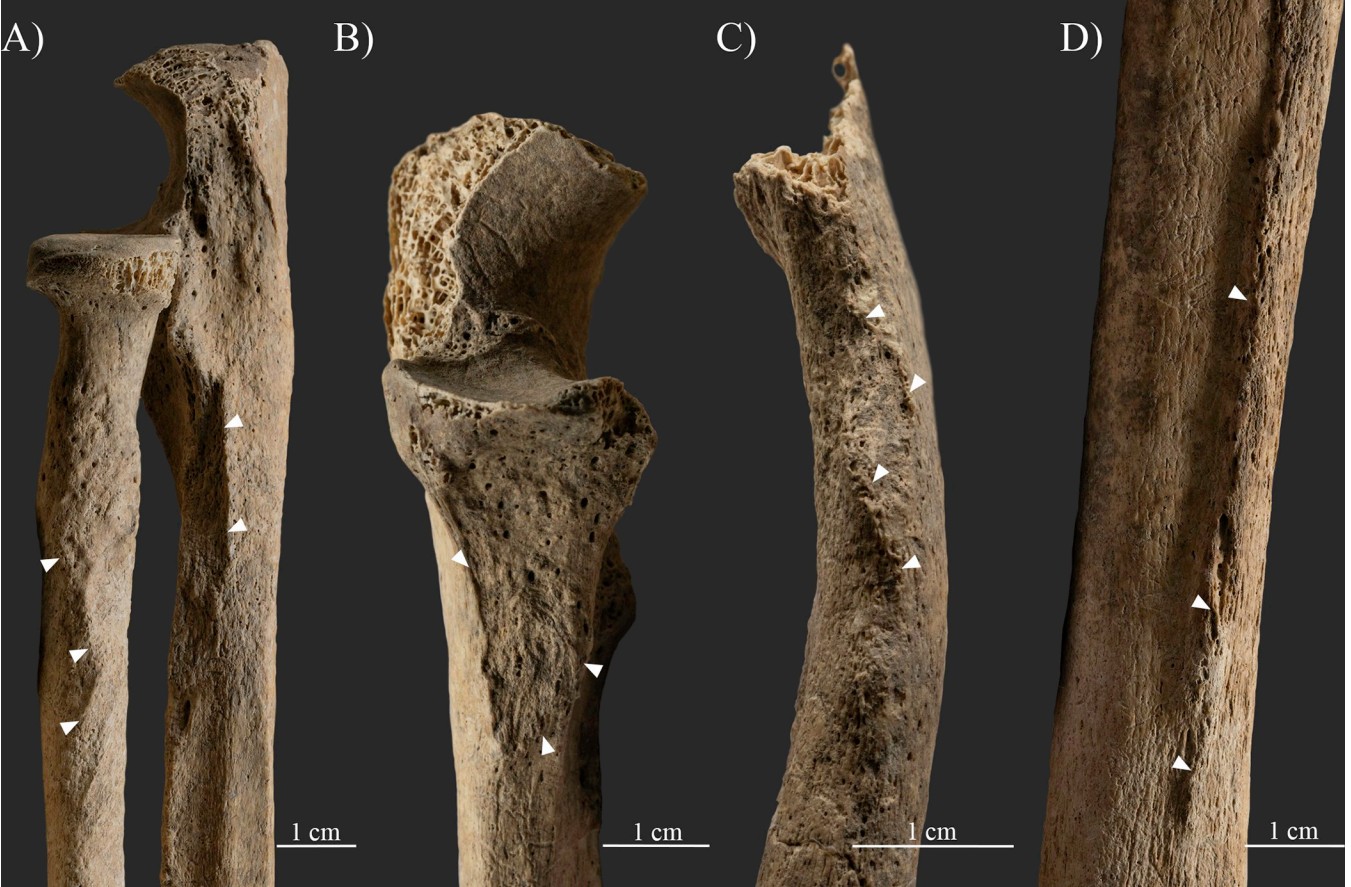

**Fig 9. Characteristic entheseal changes observed on the skeletal remains of SH-63 (photos by Luca Kis).** A) Changes present on the supinator crest of the left ulna and the lateral surface of the proximal third of the left radius (insertion sites of the supinator muscle); B) Entheseal changes on the ulnar tuberosity (insertion site of the brachialis muscle) of the left ulna; C) Entheseal changes on the rugosity for the deltoid muscle on the right clavicle; and D) Entheseal changes present on the linea aspera of the right femur (insertion site of the vastus medialis, vastus lateralis, adductor longus, adductor brevis, adductor magnus, and biceps femoris muscles).

diagnosis. Secondly, poor bone preservation complicates the detailed discussion of diagenetic changes in relation to bone loss. Nevertheless, it is unlikely that diagenetic changes alone account for the observed changes, and the presence of pathological conditions, including osteoporosis, should be considered, especially given the antemortem bone fractures found in the skeleton of SH-63. In earlier studies focusing on the paleopathological investigation of the Sárrétudvari–Hízóföld series, six cases with possible traces of osteoporosis were registered, including five adult females and one adult of indeterminate sex. In contrast, no osteoporosis-related changes were observed in adult males (e.g., [151, 180]). However, comparative analysis with currently available data at the populational level is constrained by methodological changes in paleopathological diagnostics over the subsequent decades.

Traces of traumatic lesions were detected in three different bones of the upper limb, specifically on the right humerus, right scapula, and left scapula. Although post mortem changes, such as the loss of the right humeral head, limit the observations [4, 166, 181–183], the changes described on the humerus, including the exostoses and angulation of the proximal bone end, are related to a healed antemortem trauma, particularly a possible two-part neck fracture caused by indirect trauma (e.g., [90, 166, 168, 183–185]). This type of trauma is commonly caused by incidental injuries, such as an accidental fall onto an outstretched arm, and most

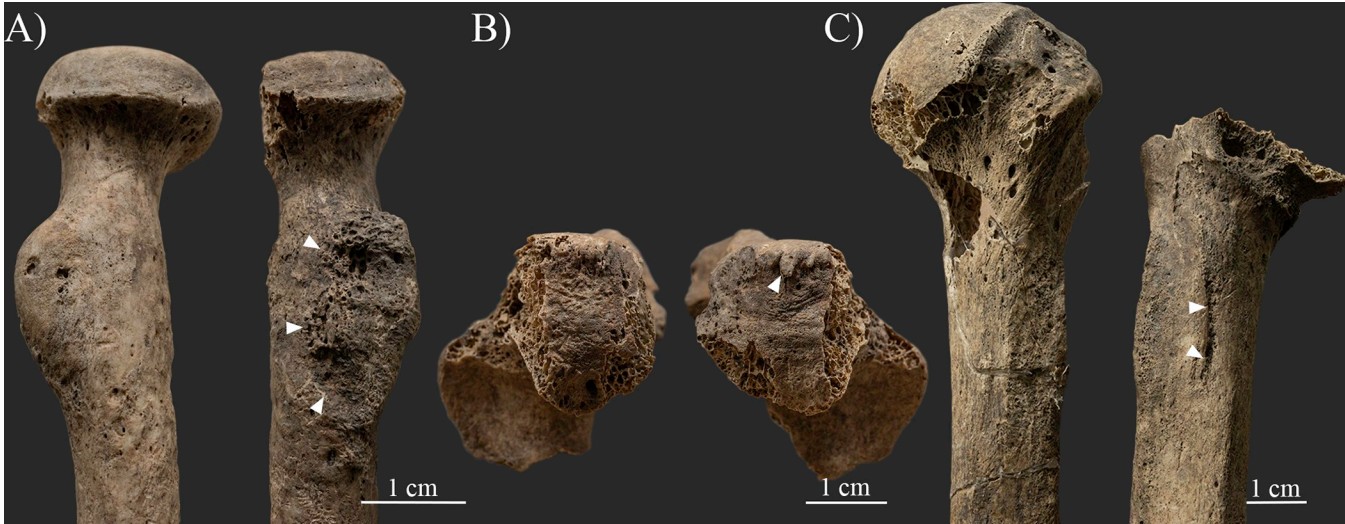

**Fig 10. Entheseal changes appearing asymmetrically between the left and right sides (photos by Luca Kis).** A) Changes observed on the radial tuberosity of the radius (insertion site of the biceps brachii muscle); B) Changes present on the olecranon of the ulna (insertion site of the triceps brachii muscle); and C) Changes documented on the crest of the lesser tubercle of the humerus (insertion site of the latissimus dorsi/teres major muscles).

often occurs in adolescents and older adults, particularly females affected by changes related to osteoporosis (e.g., [166, 168, 175, 183]). Based on the osteolytic lesions and new bone formations present on the anterior and medial parts of the bone surface, an infection during the healing process cannot be excluded. Parallel to the healed fracture line of the right humeral neck, a break in the continuity of the lateral margin of the right scapula is present. The observed bone changes can be considered as evidence of an antemortem complete fracture (e.g., [166, 168, 183]). While traces of healing are observed (e.g., rounded margins and new bone formation), the lateral margin of the scapula remained in two separate parts, forming a secondary facet between them. Based on its location and the positioning of the fracture line, it is possible that the fracture of the right scapula is related to the same injury that affected the right humeral neck. However, the precise extent and characteristics of the fracture are unknown, as the surrounding bone parts are *post mortem* missing. Fortunately, the left scapula is better preserved, and the traumatic lesions present on the bone could be more precisely observed. Based on the literature, the body of the left scapula was affected by an incidental injury, and the described symptoms are related to a healed, antemortem, comminuted fracture (e.g., [166, 168, 183]). Scapula fractures, often occurring in middle-aged and old adults [183], are considered uncommon as the overlying muscle groups usually protect the bone from injuries (e.g., [168, 183, 185]). In this case, possibly the function of the affected muscles, particularly the subscapularis muscle at the costal surface and the infraspinatus and teres minor muscles at the dorsal surface, was also altered or limited due to the deformation and fixation of the fractured bone parts in a secondary position. Nevertheless, the insertion sites of the main muscle groups of the trunk, shoulder, and arm (e.g., deltoid, latissimus dorsi, teres major, triceps brachii, and biceps brachii muscles) were not affected, and presumably these muscles were still functioning. The location and characteristics of the fractures observed on SH-63 suggest that the injuries were the result of at least two separate incidental traumas. However, the time that elapsed between the occurrence of the injuries is unknown, as all three fractures have undergone a healing process, but none of them were properly healed. Certainly, the general state of the bones and changes related to osteoporosis are likely to be the main factors responsible for the severity of the traumatic lesions, specifically, the relatively high number of

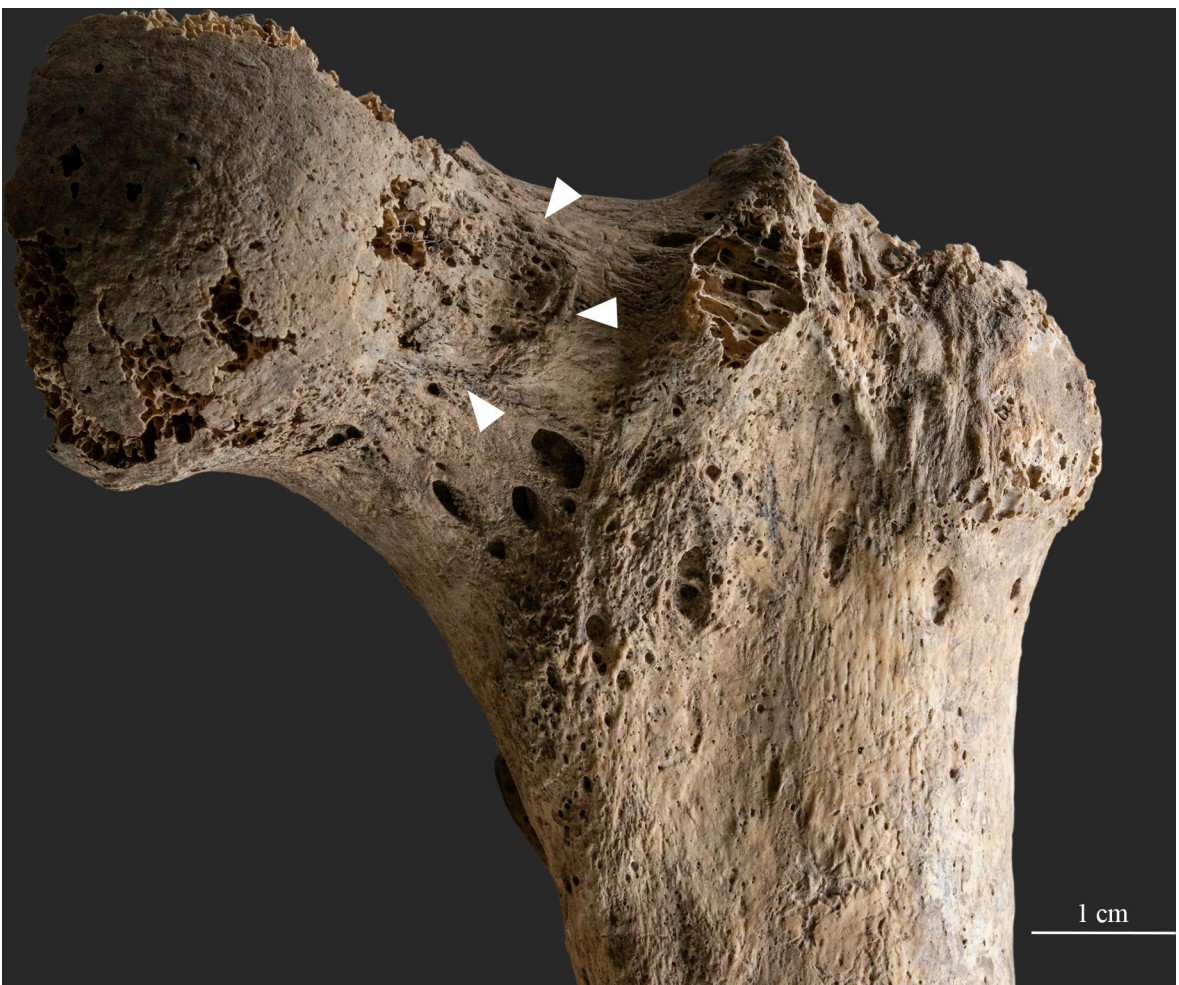

**Fig 11. The imprint with a slightly rough surface and a bony rim observed on the anterior margin of the left femoral neck (photo by Luca Kis).**

antemortem fractures present on the skeleton (e.g., [166, 168, 175, 183]). On the other hand, it is likely that other factors, particularly the lifestyle, also increased the risk of suffering injuries. All three traumas of the upper limb bones of SH-63 were caused by incidental traumas, possibly falling directly onto the shoulder and/or the outstretched arm. Traumas and bone fractures are also known in other skeletons from the Sárrétudvari–Hízóföld series (e.g., [144, 145, 155, 158]). Previous paleopathological studies of the series have highlighted that while fractures and other traumas are common among males, such changes are exceptionally rare in females, suggesting differences in lifestyle between the sexes (e.g., [78, 151, 186]). Particularly, polytraumatism and fractures of the upper limb occurred in the highest number in the group of individuals buried with weapons and/or horse riding equipment, suggesting that lifestyle–including horse riding–is responsible for the high number of traumas in the Sárrétudvari–Hízóföld series (e.g., [144, 145]). In this regard, the traumas detected on SH-63 are consistent with the injuries observed on individuals buried with weapons and/or horse riding equipment from the Sárrétudvari–Hízóföld series.

Joint changes and entheseal changes were observed on the preserved bone elements in a relatively high number. However, it must be acknowledged that the evaluation of the described

bony changes is limited, as non-mechanical factors (e.g., genetics, sex, age, and metabolic disorders) can also influence their generalized development (e.g., [72, 88–93]). Thus, the possible advanced age of the individual and the traces of a metabolic disorder (osteoporosis) on the bones might obscure the changes related to physical activities. Additionally, in most cases, bone preservation did not allow the registration of the true prevalence of the changes, hindering the reconstruction of the possible movements and physical activities. Comparative analysis is also limited, as this is the first known female burial with weapon equipment from the 10th-century-CE Carpathian Basin. Previous studies on paleopathological and supposed activity-related skeletal changes in the series have revealed a disparity between the sexes in the distribution of these alterations, with entheseal and joint changes predominantly found in males, indicating differences in lifestyle [78, 151, 180]. For instance, entheseal changes in the upper limb bones were reported in 21 males but only in one female (e.g., [180]). However, comparative analysis with earlier findings is limited due to methodological changes in the registration and evaluation of activity-related changes over the subsequent decades.

Nevertheless, we found asymmetry in the occurrence and characteristics of entheseal changes at the insertion sites of the main muscles of the upper limb bones, suggesting the influence of physical stress on their development. In these cases, the changes were more pronounced on the right side. These muscles are responsible for complex movements, particularly the extension (triceps brachii muscle) and flexion (biceps brachii muscle) of the forearm, inner rotation/adduction of the arm (teres major muscle), as well as pulling the arm downward and backward when the trunk is fixed (latissimus dorsi muscle) [187, 188]. Although no entheseal or joint changes specific to a particular activity were found on the skeletal remains, the observed joint and entheseal changes in SH-63 (S1 and S2 Tables) were also described in other individuals from the Sárétudvari–Hízóföld series buried with horse riding-related deposits and/or weapons (e.g., [59, 144, 158]).

The bone changes registered on the femoral neck can be considered as a probable anteroiliac plaque [189]. Morphological variants of the femoral neck have long been associated with activities, such as squatting, running, walking downhill, sitting cross-legged, and horse riding [189]. However, more recent studies have highlighted that their interpretation is still problematic, and features such as the plaque might even be considered as normal conditions of the femur (e.g., [189, 190]). Among others, femoroacetabular impingement caused by a repetitive contact between the acetabular rim and the femoral head-neck junction might be of great importance in understanding these variations [190]. This condition is thought to be the cause of hip osteoarthritis in many cases (e.g., [190, 191]). In this regard, it is possible that the features on the femoral neck and the joint changes observed on the acetabular rim of SH-63 are associated with and related to the forced movements of the hip.

In conclusion, numerous changes in the musculoskeletal system have been observed on the bones of SH-63 (Fig 12).

The interpretation is highly limited due to the poor state of preservation of the bones, the possible advanced age of the individual, and the multifactorial etiology of the registered changes. While advanced age and associated degenerative processes could contribute to the observed changes, it is essential to consider that repetitive physical stress and lifestyle factors also play a significant role. The presence of specific alterations and asymmetry at certain entheseal sites suggests that physical stress related to the individual's lifestyle remains a key factor in these changes. These changes, including entheseal changes, joint changes, morphological variants, and traumas, are not specific to any particular activities. Additionally, it is possible that the activities the individual could have practiced on a daily basis were limited due to the current changes present on the bones (e.g., improperly healed fractures and severe joint changes). This phenomenon is not unique to the Sárrétudvari–Hízóföld series. Other cases have been

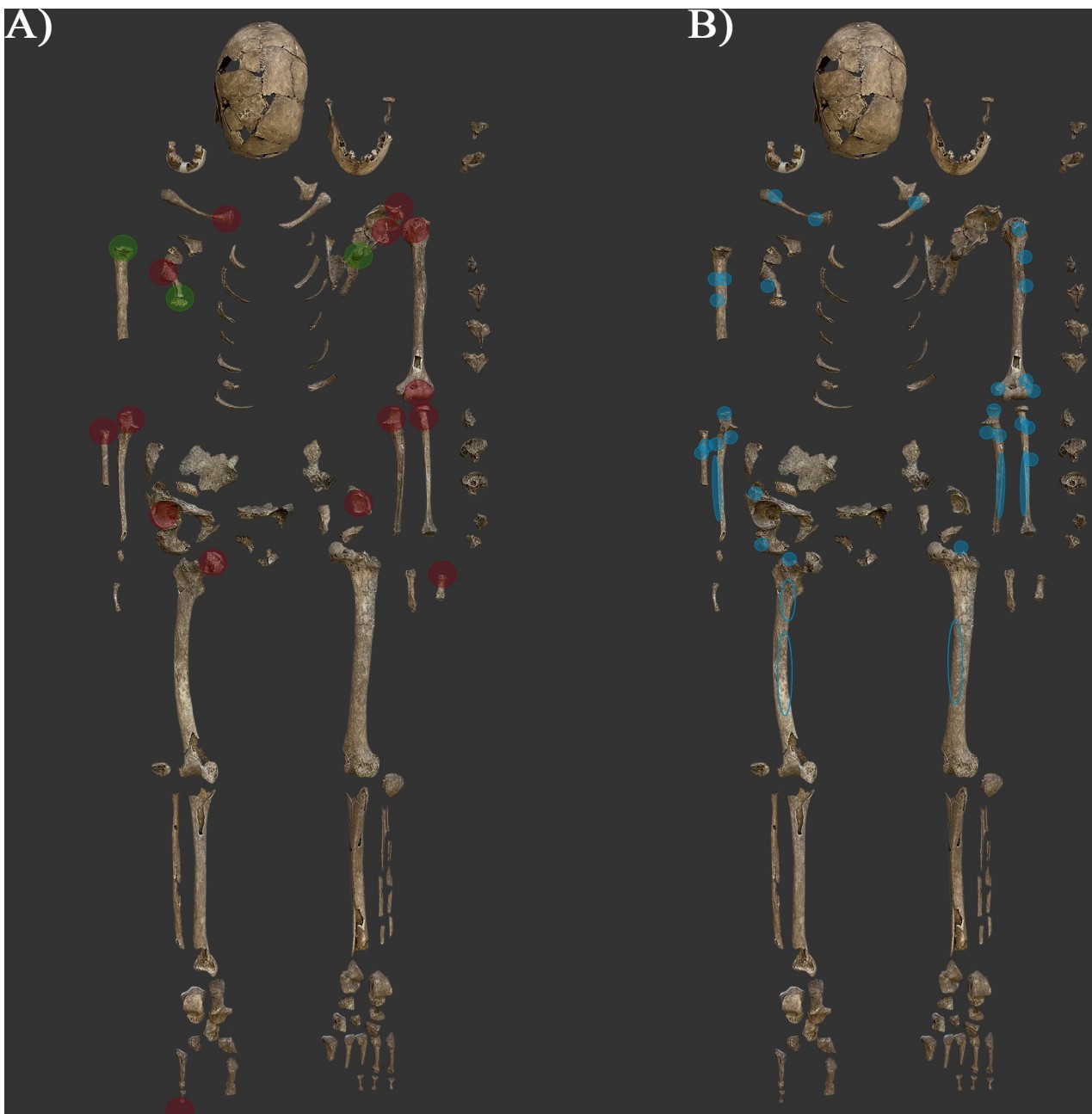

**Fig 12. Changes in the musculoskeletal system observed on the extant bones (photos by Luca Kis).** A) Distribution of observed traumatic lesions (green) and joint changes (red); and B) Distribution of observed entheseal changes (blue). Notes: blue circles without filling on the femurs indicate entheses located on the posterior part of the bones. A detailed list of the observed entheseal and joint changes is provided in S1 and S2 Tables.

documented where the burial contained weapons, yet sever pathological changes were present in the bones, potentially limiting the use of these weapons on a daily basis. For instance, burial No. 183 contained archery equipment and horse riding-related deposit [149], but the skeletal remains exhibited extensive degenerative joint changes, including eburnation, contour deformation, and surface remodeling present on the bone surfaces of both elbows (e.g., [155, 157]). Nevertheless, the observed patterns of the bone changes in SH-63 are consistent with features

found in the skeletal remains of individuals buried with weapons and/or horse riding equipment in the Sárrétudvari–Hízóföld series suggesting similarities in the lifestyle of these individuals.

## Evaluation of the burial from the perspectives of funerary archaeology

In the examination of the burial customs, it was observed that the orientation of grave No. 63 slightly deviated from the general west-east direction, displaying a minor variation toward the south-north direction [149]. This phenomenon is not uncommon, and similar tendencies have been noted in burials along the western border and other sections of the cemetery. Moreover, this particular grave orientation has been documented in other burials featuring weapon equipment, such as grave No. 66 in the southern part of the cemetery, which included archery equipment and a saber [149].

The positioning of SH-63 in the grave is unusual within the 10th-century-CE archaeological horizon of the Carpathian Basin [192]. Only one other case, the skeleton from burial No. 69, was found in a slightly flexed position with bent knees in the Sárrétudvari–Hízóföld series. Previous studies indicated that the skeletal remains from grave No. 69 belonged to a female individual (e.g., [150, 151]), buried with a silver penannular hair ring, two bronze finger rings, and a knife [149]. It was previously suggested that this particular body position was more prevalent in poorly furnished burials or even graves of presumed "slaves" [192]. However, to the best of our knowledge, no systematic investigation has been conducted to analyze the archaeological and bioarchaeological background of individuals buried in a flexed position during this period. In case of SH-63, the number and composition of the grave goods do not support the idea that social differences account for the body position in the grave. The association with the biological sex of the deceased, on the other hand, cannot be conclusively ruled out, but with the current available data, it remains a controversial issue.

The grave goods consist of artifact types commonly found in 10th-century-CE burials in the Carpathian Basin. The inventory is relatively simple (e.g., lack of clothing ornaments and finely-crafted jewelry), a common observation in the Sárrétudvari–Hízóföld cemetery, especially in the case of male burials. In addition, these items were found in their expected position, in accordance with their presumed functions. Penannular hair rings are the most common type of jewelry found in burials irrespective of the biological sex or potential social and economic status of the deceased, both in the Sárrétudvari–Hízóföld burial ground and in other 10th-century-CE cemeteries of the Carpathian Basin (e.g., [116, 141, 149, 193, 194]). Bell buttons have also been discovered in male and female burials, regardless of the assumed social and economic status of the individual [193, 195]. However, these clothing elements were more frequently found in female burials in the Sárrétudvari–Hízóföld cemetery [149, 195]. Beads are one of the artifact types that were traditionally associated with females and sub-adults. However, it is known that beads might also appear in 10th-century-CE male burials. Differences were observed in their number and ornamentation between the two sexes, as male burials contained only monochrome or eyed beads, and their number on the string did not exceed five [103]. Similar tendencies were noted in the Sárrétudvari–Hízóföld cemetery, where besides two adult males with a single bead, strings of beads were found in burials of anthropologically confirmed sub-adults and adult female individuals (e.g., [75, 144, 149–151]). Specifically, the burial of SH-63 contained a string of beads with 14 beads in different types and colors, showing analogies with other female burials from the cemetery featuring string of beads [149].

The identifiable arrowhead found in the grave belongs to the so-called armor-piercing type [149]. Although this type of arrowhead is known in the 10th-century-CE cemeteries of the Carpathian Basin, it appeared rarely compared to the more common deltoid-shaped forms (e.g.,

[116, 133]). This type of arrowhead was also found in other graves in the Sárrétudvari–Hízó-föld cemetery, equipped with a quiver and/or bow.

The characteristics and precise structure of the quiver found in the grave remain unknown, as the poor preservation allowed only the registration of its presence. Based on the described fragments [149], it can be assumed that the quiver contained structural elements commonly found in quivers form the Sárrétudvari–Hízóföld series and other 10th-century-CE cemeteries in the Carpathian Basin (e.g., [116]).

The antler plate found near the hip belonged to a composite bow, covering the lateral part of the grip. Plates with similar morphology have also been found in other graves of the Sárré-tudvari–Hízóföld cemetery (e.g., grave Nos. 3, 185, and 251) [149]. The function of these plates on the bows is not yet clear as they cannot be considered obligatory structural elements. While the number of burials with bow plates exceeds 300 (e.g., [118, 131, 132]), their ratio is still significantly less compared to the total number of 10th-century-CE burials with other elements of archery equipment (e.g., quiver or arrowhead). In addition to the grip, plates covering the tips of the bow–predominantly the lateral sides–are also known from 10th-century-CE burials [133]. The number of the antler plates on the 10th-century-CE "Magyar" type bows varies between 1 and 6, but 1–2 plates have been more commonly found in the burials than the full set of them (2 on the grip and 2–2 on the tips) (e.g., [118]). Other burials with only one plate, covering the bow grip, are also known in the Sárrétudvari–Hízóföld cemetery, such as grave No. 66, which contained arrowheads and a saber, as well [149]. In this regard, the bow remain found in grave No. 63 does not represent an extraordinary case but fits well with the tendencies observed in the Sárrétudvari–Hízóföld cemetery and, in general, the 10th-century-CE archaeo-logical material. While the origin and symbolic significance of these artifacts (e.g., personal items or gifts) remain unclear, the location of the bow remain in the grave suggests a connection with the deceased, as the plate was in a rather functional position, as if 'she was holding it in her left hand', such as described by the archaeologist in charge of the excavation of the cemetery [149].

In summary, the burial contained grave goods typical of both sexes, with jewelry usually associated with females and weapon associated with males. The weapon, particularly archery equipment, exhibit characteristics similar to related artifacts from other burials. Apart from the positioning of the deceased, the burial customs and grave goods documented in the grave of SH-63 show close analogies with other burials with weapons in the Sárrétudvari–Hízóföld cemetery.

## Conclusions

In this paper, we presented the results of our interdisciplinary research, which was conducted on the archaeological and anthropological material deriving from grave No. 63 from the 10th-century-CE cemetery of Sárrétudvari–Hízóföld. This unique burial contained jewelry traditionally associated with females and weapon typically linked to males. While instances of male burials with jewelry are documented from the era (e.g., [103]), the burial of a female with weapons, including full archery equipment or melee weapons, has not previously been identified in 10th-century-CE cemeteries of the Carpathian Basin.

We analyzed the skeletal remains (SH-63) using both anthropological and archaeogenetic methods. The results consistently indicated that the skeleton belonged to an adult female. The examination of the burial customs and grave goods revealed that no significant difference can be detected on the archaeological level between SH-63 and males buried with weapons in the Sárrétudvari–Hízóföld cemetery. It is unlikely that weapon was provided in her grave as amulet; rather, common factors, such as social and economic status, as well as possible lifestyle could have influenced the applied burial customs and composition of the grave goods.

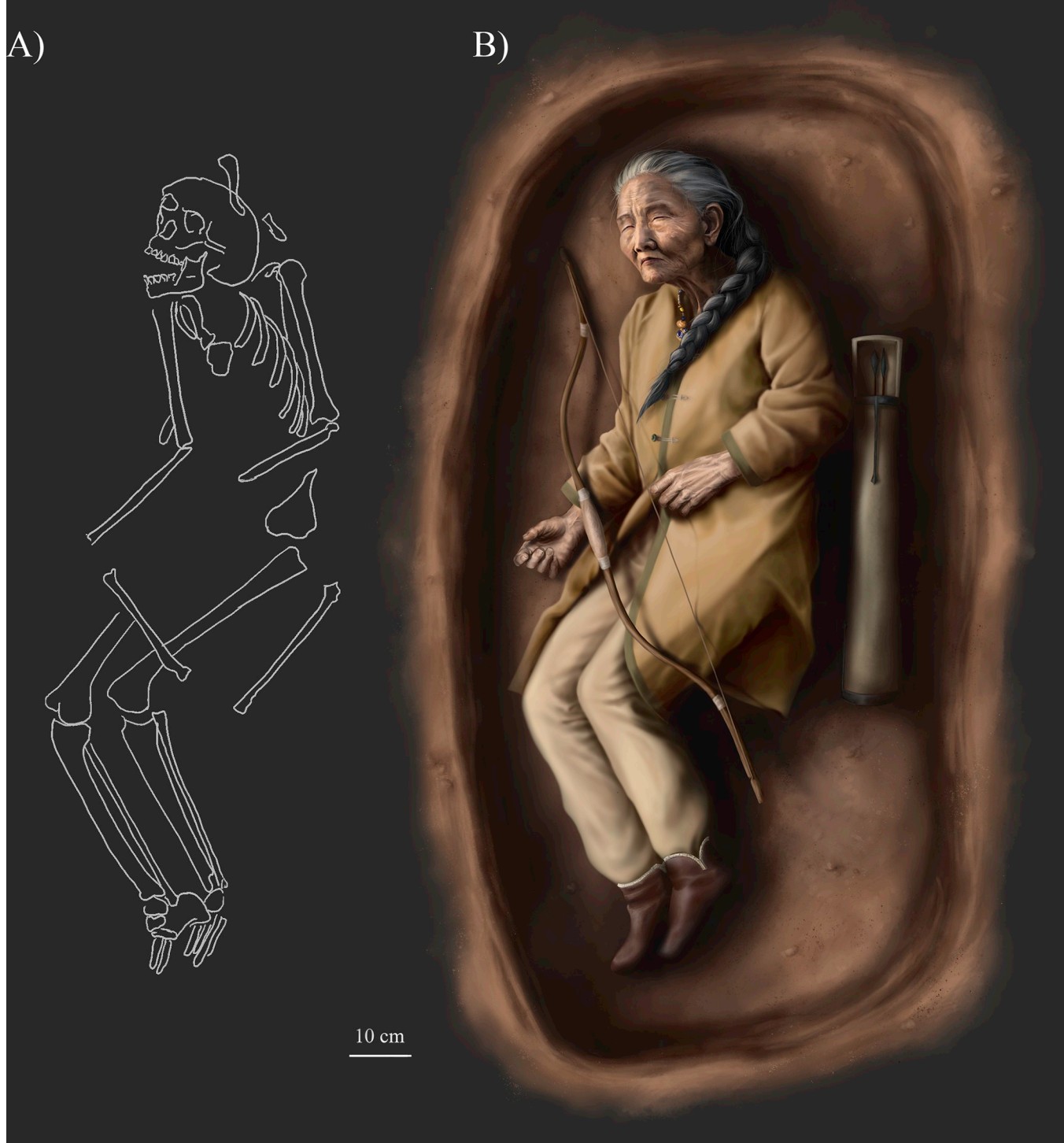

**Fig 13.** A) Silhouette of the skeleton of SH-63 (drawing by Luca Kis based on the original field drawing by Ibolya M. Nepper); B) Illustration of the burial based on archaeological, anthropological, and archaeogenetic data (drawing by Luca Kis).

However, evaluating the possible "occupation" of the individual is a complex problem. Traces of pathological disorders and changes supposedly related to physical activities were observed on the skeletal remains, but none of the registered changes were specific to a certain activity such as horse-riding or archery. Additionally, it is possible that the activities she could have practiced on a daily basis were limited due to the current changes present on the bones. Nevertheless, similarities can be found in the pattern of these changes between SH-63 and other individuals buried with weapons in the Sárrétudvari–Hízóföld series, which could have resulted from practicing similar activities during their life.

Certainly, one of the most intriguing questions is whether the case can be considered a warrior burial. Unfortunately, at the current research level, this must remain an open debate. First, the term "warrior" involves specific aspects on social and legal levels for which sources other than archaeological and anthropological are also required for identification, but no written data are available concerning warrior women among the Magyars in the 10[th] century CE. Additionally, the origin and symbolic significance of the artifacts—whether they were the personal items of the deceased or gifts from the community—remain unclear. Furthermore, in nomadic tribes of the eastern steppes, similar to the early Magyars, it was common for females to learn how to defend themselves and the livestock to survive [142, 143]. While this probably resulted in practicing similar daily activities with males, they were not necessarily considered as dedicated warriors.

The main limitation of the interpretation lies in the uniqueness of SH-63 and the current lack of comparative data. Nevertheless, we can confidently conclude that this individual indeed represents the first known female burial with weapon from the Hungarian Conquest period in the Carpathian Basin (Fig 13). Moreover, SH-63 may have had a lifestyle similar to other individuals buried with weapons in the cemetery. The Sárrétudvari–Hízóföld series, and particularly this burial, have the potential to enhance our understandings of the social organization of the Hungarians in the 10[th] century CE. Most of the burials equipped with weapons from this period are still associated with males and a significant increase in the number of similar female cases is not expected. However, this case highlights the methodological issues concerning the interpretation of data on sex and lifestyle, which are frequently overinterpreted in related archaeological and anthropological studies.

## Supporting information

**S1 Table. Description of joint changes observed on the extant skeletal remains of SH-63.**
(DOCX)

**S2 Table. Description of entheseal changes observed on the extant skeletal remains of SH-63.**
(DOCX)

**S1 Text. Detailed description of archeogenetic analysis of SH-63.**
(DOCX)

**S1 File. PMD stats of SH-63.** PMD summary and PMD counts received from the analysis of the petrous bone sample.
(XLSX)

## Acknowledgments

We extend our sincere gratitude to Dr. János Dani, Barbara Kolozsi, Dr. Ibolya M. Nepper, and Zoltán Faur (Déri Museum, Debrecen, Hungary) for their permission and invaluable

assistance in studying the archaeological material and documentation of the Sárrétudvari–Hízóföld cemetery. We are particularly grateful to Zoltán Faur for the photos of the artifacts found in grave No. 63. We would also like to express our appreciation to our colleagues at the Department of Biological Anthropology, University of Szeged, for their valuable support throughout this study.

## Author Contributions

**Conceptualization:** Luca Kis, William Berthon.

**Data curation:** Balázs Tihanyi, Kitti Maár, Luca Kis, Zoltán Maróti.

**Formal analysis:** Zoltán Maróti.

**Funding acquisition:** Balázs Tihanyi, Luca Kis, Tibor Török.

**Investigation:** Balázs Tihanyi, Kitti Maár, Alexandra Gînguă, Gergely I. B. Varga, Bence Kovács, Oszkár Schütz, Zoltán Maróti, William Berthon.

**Methodology:** Balázs Tihanyi, Kitti Maár, Alexandra Gînguă, Gergely I. B. Varga, Bence Kovács, Oszkár Schütz, Endre Neparáczki, Olga Spekker, Zoltán Maróti, William Berthon.

**Project administration:** Balázs Tihanyi.

**Resources:** György Pálfi, Endre Neparáczki, Tibor Török.

**Supervision:** György Pálfi, Endre Neparáczki, Tibor Török, Olga Spekker, Zoltán Maróti, William Berthon.

**Visualization:** Luca Kis.

**Writing – original draft:** Balázs Tihanyi, Kitti Maár, Zoltán Maróti.

**Writing – review & editing:** Balázs Tihanyi, Luca Kis, Olga Spekker, William Berthon.

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
