## [Decision Letter · Decision Letter 0]

1 Aug 2024

PONE-D-24-19948‘But no living man am I’: Bioarchaeological evaluation of the first-known female burial with weapons from the 10th-century-CE Carpathian BasinPLOS ONE

Dear Dr. Tihanyi,

Thank you for submitting your manuscript to PLOS ONE. After careful consideration, we feel that it has merit but does not fully meet PLOS ONE’s publication criteria as it currently stands. Therefore, we invite you to submit a revised version of the manuscript that addresses the points raised during the review process.

The manuscript entitled 'But no living man am I': Bioarchaeological evaluation of the first-known female burial with weapons from the 10th-century CE Carpathian Basin' presents a case of a female burial with a set of objects referable to weapons together with fairly typical grave goods for a female grave.

The three reviewers welcomed the manuscript with positive comments but also expressed critical issues that need to be addressed by the authors.

In my view, there are two general problems with this contribution:

1.    The authors commendably take a critical approach to the data and emphasise how other explanations may be valid in interpreting the funerary/anthropological/paleopathological context. Converselym, this caution is not matched by the same degree of terminological prudence (as one reviewer notes the presence of an arrowhead and a single bow plate do not constitute sufficient evidence of a female burial with weapons) and the title should be changed, especially by removing the quotation from Tolkien that refers directly to a female warrior character.

2.    The authors should move the lengthy description of genetic analyses to the supplementary information and reduce the lengthy descriptions of possible methodologies (see those for sex determination). At the same time, more effective and less time-consuming techniques for proteomics of dental enamel are ignored, just as teeth are never analysed.

We look forward to receiving your revised manuscript.

Kind regards,

Luca Bondioli, PH.D.

Academic Editor

PLOS ONE

2. We note that Figures 1, 2, 3, 4, 5, 6, 7, 8, 9, 10, 11, and 12 in your submission contain [map/satellite] images which may be copyrighted. All PLOS content is published under the Creative Commons Attribution License (CC BY 4.0), which means that the manuscript, images, and Supporting Information files will be freely available online, and any third party is permitted to access, download, copy, distribute, and use these materials in any way, even commercially, with proper attribution. For these reasons, we cannot publish previously copyrighted maps or satellite images created using proprietary data, such as Google software (Google Maps, Street View, and Earth). For more information, see our copyright guidelines: http://journals.plos.org/plosone/s/licenses-and-copyright.

1. You may seek permission from the original copyright holder of  Figures 1, 2, 3, 4, 5, 6, 7, 8, 9, 10, 11, to publish the content specifically under the CC BY 4.0 license.  

Additional Editor Comments (if provided):

Reviewers' comments:

Reviewer's Responses to Questions

**Comments to the Author**

1. Is the manuscript technically sound, and do the data support the conclusions?

Reviewer #1: Yes

Reviewer #2: Yes

Reviewer #3: Yes

2. Has the statistical analysis been performed appropriately and rigorously? 

Reviewer #1: I Don't Know

Reviewer #2: N/A

Reviewer #3: N/A

3. Have the authors made all data underlying the findings in their manuscript fully available?

Reviewer #1: Yes

Reviewer #2: Yes

Reviewer #3: Yes

4. Is the manuscript presented in an intelligible fashion and written in standard English?

Reviewer #1: Yes

Reviewer #2: Yes

Reviewer #3: Yes

5. Review Comments to the Author

Reviewer #1: The research paper, entitled "But no living man am I": Bioarchaeological evaluation of the first-known female burial with weapons from the 10th-century CE Carpathian Basin" by Balázs Tihanyi and co-authors presents the bioarchaeological study of a female burial (No 63), discovered within the 10th century CE century CE cemetery of Sárrétudvari-Hízóföld (eastern Hungary).

The tomb yielded a grave goods set that is related to both the female sphere (including ornaments that recurred in other female and subadult tombs at the site) and the male sphere (including weapons and an archery set, which were typical of the Sárrétudvari-Hízóföld male tombs).

The study of this funerary context addresses a complex and contentious topic: the identification of female warrior status in past human societies through the examination of funerary evidence. Furthermore, it considers the question of determining an individual's sex and gender based on biological parameters and material culture.

The authors are fully aware of the critical nature of the topic and I was pleased to see a lengthy paragraph in the introduction to the paper outlining the state of the art of approaches to the study of gender identity and sex determination from an archaeological and bioarchaeological perspective.

In general, this article is well written, the English used is fluid and the approach to the scientific data is critical.

In the presentation of the 'palaeopathological' data and subsequent discussions, the authors adopt a nuanced approach, acknowledging the limitations of the sample and the multi-etiology of the lesions under consideration.

However, I would recommend that the authors consider more explicitly that, with the exception of the fractures at the humeroscapular joint, the observed lesions may be more likely to be related to the mature age of the individual and not to a specific activity associated with.

Moreover, the presence of the archer's set in the tomb (an element that, in a post-processualist perspective, could also be indicative of a 'gift' or the exaltation of a status and not of the deposition of the individual personal objects) is worthy of further consideration.

Additionally, as proposed below, I would like the authors to extend the comparison between the bioarchaeological evidence gathered in this study on individual SH 63 and that available not only on the other individuals with weapons in the tombs, but also on the other female individuals and, in general, for the necropolis as a whole.

This would help to avoid forcing a connection (tombs with weapons) that could produce flawed observations.

From a methodological and analytical perspective, it is my contention that the application of further investigative tools, such as bone histology, would enhance the reliability of the conclusions drawn. This would entail the verification of phenomena such as osteopenia, osteoporosis and bone fragility, and the distinction between these and phenomena of diagenetic bioerosion. Furthermore, radiographic examination of the fracture injuries would enable a more detailed investigation of the state of bone remodelling and healing.

It would also be of interest to ascertain whether there are any differences between the sexes and between individuals of the same sex in terms of nutrition (carbon and nitrogen isotopes) and origin (oxygen and strontium isotopes, for example).

Finally, I would recommend to provide images of a higher quality, as the low resolution of the figures submitted for review made it challenging to accurately compare the lesions described in the text with their photographic documentation.

Below are a few more detailed observations:

lines 47-49: Although the discussion makes it clear that the hypotheses regarding the presence of the fractures and other bone lesions (the latter being of a multi-etiological nature) may be connected either to the activities carried out in life by the woman (use of the bow, horse-riding) or to other idiosyncratic and/or ontological factors, the abstract places too much emphasis on seeking a connection between the evidence found on the remains of the female with weapons and the other armed males in the necropolis. This connection may lead to a flawed interpretation of the data, which has not been compared with the other unarmed individuals in the necropolis, both male and female.

It would be advisable for the authors to consider making this sentence more nuanced.

lines 61-62: When referring to archaeology, I would always use the term gender and not sex.

Additionally, the expression 'classic archaeology' can be confused with 'classical', whereas the authors here refer to 'traditional archaeology'.

lines 66-67: The term 'classical anthropology' should be changed to 'traditional anthropology'.

It would be advisable to revise the sentence, as the methods of traditional anthropology, based on morphology and morphometrics, although less expensive than DNA techniques, are no simpler.

line 84: Please, add to the bibliography on the use of amelogenin in archaeological contexts the article by Lugli and colleagues (2019) ' Enamel peptides reveal the sex of the Late Antique 'Lovers of Modena'.

linee 219-221: Please, provide a more detailed description of the composition of the sample and how the number of males and females has changed since the previous analyses. Out of a total of 162 adults, the previous analyses identified 70 females and 85 males (155 sexually determined and 7 non-determined), while the sex was subsequently confirmed for 52 females and 69 males. Has the number of undetermined people therefore increased?

line 259: Please, provide a rationale for the authors' determination of the percentage of exhibit preservation as greater than 50 per cent, and include a reference.

line 263: As above. Please, add reference.

line 269: Please, add reference for thickness of cranial bones.

line 272: Please, add reference for age estimation based on auricular surface or symphysial surface.

Furthermore, the authors do not discuss the estimation of age at death based on the wear of the occlusal surfaces of the dental enamel. It is unclear why this parameter was not considered for the individual SH63.

lines 357-364: I would move the ethical statement to the appropriate space in the submission form.

linee 404-409: Please, add reference for all the changes in bone anatomy described in this paragraph.

lines 418-422: Please, provide a detailed account of the methodology employed by the authors to distinguish this type of lesion from post-depositional and diagenetic alterations. Additionally, please include a comprehensive bibliography. As previously stated, these types of alterations were deemed worthy of investigation through the analysis of bone cortical histology.

line 541: Please, add the reference to ' by Skoglund and colleagues'.

lines 633-637: Although the authors acknowledge the multi-etiology of the lesions identified on the remains of SH 63, they do not explicitly address the possibility that these lesions may be caused by physiological and degenerative-ontological processes within the skeleton, rather than being strictly related to the lifestyle of the individual. The authors do not provide a precise age at death for the individual, although they do hypothesise osteopenia and osteoporosis (post-reproductive age?).

It would be advisable for the authors to consider the influence of the individual's age at death when expanding this part of the discussion.

This factor should also be considered when comparing injury data from other individuals who were armed. Rather than considering similarities in lifestyle, can we posit that they were compatible individuals in terms of age at death?

Reviewer #2: Well written article on an interesting burial context from 10th century AD Hungary. This article is presented as the ‘first known female burial with weapons (plural). The authors describe the remains in a lot of detail, and are careful to avoid overinterpretation, and clearly indicate where the problems are related to preservation and context. In general I find the description well done and the discussion well rounded, and the conclusion convincing.

I have a few notes for nuances.

As indicated in the article, multiple burials are known from this time period where women were buried accompanied by arrowheads. They have only been interpreted differently by previous scholars (as amulets), due to the lack of additional gear (and likely a different mentality in interpretation in previous times). The presence of the bow plate in combination with the arrowhead and possibly quiver remnants is a good indication that the arrowhead would be a utility item/weapon rather older interpretations for the arrowheads. However, this is the only weapon found in the grave, so it is not plural, and if other graves also contain arrowheads, is it really the first? You could consider rephrasing 'first ...with archery gear'

As the authors themselves also indicate in the discussion, archery equipment is likely to have been part of a toolkit that allowed also for defense, and other indications are also present for women in this society to have been skilled horseriders as there have also been numerous indications of women from this time period being buried with horse gear. Therefore, horse-riding and using bow and arrow is definitely a possible component of the female lifestyle for this time period and population. It is however absolutely the case, as the authors describe, that this seems to be a rarer occasion where the skeletal remains can be interpreted with confidence on being female, and have a combination of a bow plate and an arrowhead. This in combination with the healed pathological descriptions on the women having had an active lifestyle, is to me convincing.

-Small terminology note: The authors refer to the bows of this timeperiod as compound bow. I think this is a translation issue, as it should be composite bow. The term "compound bow" is more commonly used to refer to specific type of modern bow that has a system of pulleys and cables that help in drawing and holding the bowstring. The type of bows discussed in this article should be referred to as "composite bows."

Pathology notes: it is described that ‘the most striking feature’ in the skeleton is the ‘significant lower bone density’. If it is significant that means its measurable, but I don’t see described how this was determined, other than ‘thinning of the bone’ which is quite vague. I am commenting on this because the authors stated themselves that the bone preservation was poor. So how can this be ascertained? It is described that the bone was thin in comparison to other individuals in the series, but no data is reported for the other individuals so it is difficult to say that this is ‘significant’. Specifically in the discussion, it is stated ‘ the presence of osteoporosis…supports the results’. But there are no methods described how osteoporosis is diagnosed. Diagenetic changes in relation to bone loss for investigating age related bone loss, so the clear emphasis on the poor preservation of the bones in relation to sex determination, but no comment on this for bone density, I find a bit concerning.

I am giving a few suggestions where methodologies are described how osteopenia or osteoporosis can be measured and described using quantitative methods.

Brickley, M. B., & Agarwal, S. C. (2003). Techniques for the investigation of age-related bone loss and osteoporosis in archaeological bone. In Bone loss and osteoporosis: an anthropological perspective (pp. 157-172). Boston, MA: Springer US.

Van Spelde, A. M., Schroeder, H., Kjellström, A., & Lidén, K. (2021). Approaches to osteoporosis in paleopathology: How did methodology shape bone loss research?. International Journal of Paleopathology, 33, 245-257.

Interpretation of the trauma: I find there is some conflicting in the way the authors argue what caused the injuries. On the one hand, it is described that the type of injuries are likely/easily related to osteoporosis, which would be related to aging and bone loss. On the other hand, they interpret it as related to an active lifestyle and that the traumas are consistent with injuries observed on individuals buried with weapons and/or horse riding gear. Are these two interpretations not conflicting?

I do appreciate the authors writing from line 633 onwards that the interpretation is problematic and there is no over interpretation indeed.

Final note – I really appreciate how the conclusion is written up, as the authors clearly avoid over interpretation but at the same time indicate their findings clearly. Its interesting that this research is part of a larger study on DNA from the region – I’ll be looking forward to reading more about the results of the larger studies that can improve the context of this find.

Reviewer #3: The paper under review presents a female burial with weapons from the Carpathian Basin. This is a very interesting finding, given extensive discussions regarding the association between sex and gender. The paper, thus, makes a significant contribution and is worthy of publication, though I have some minor recommendations, as follows:

1. In the second paragraph of the Materials and Methods, the authors state that the material from the cemetery under study was initially sexed and then the analyses were repeated using different sexing methods and many of the original sex estimates were confirmed. Were the remaining original males and females deemed of indeterminate sex after re-assessment or were they classified are belonging to the opposite sex from the one originally estimated?

2. The section "Anthropological re-examination of the skeletal remains" is currently too vague; the readers have to check long lists of references to understand what methods the authors used and even then, it is not clear which methods (e.g. for sex estimation) were prioritized over others in cases where different methods produced contradictory results, or how different methods for mechanical stress or pathology were practically implemented in this case give n the partial preservation of the bones (e.g. what proportion of a bone had to be present in order for a condition to be recorded as present/absent?)

3. Why did the authors analyze three different elements (petrous bone, tooth, humerus) for aDNA? As expected, they found that the petrous bone performs much better than the others (which in this case did not work at all), so why engage in additional and unnecessary destructive analyses?

4. In the results of the mechanical stress markers, the authors provide very detailed descriptions of the trauma, entheseal changes and arthritis. However, as a result, the reader gets rather lost. It would be helpful to add a skeleton silhouette marking the joints, entheses and other anatomical areas that manifest signs of mechanical stress (e.g. use red for arthritis, blue for entheseal changes, green for trauma). This will help illustrate the distribution of relevant markers.

Once the above minor issues are addressed/explained, I am happy to see the paper published. It is an original and well contextualized study.

6. PLOS authors have the option to publish the peer review history of their article (what does this mean?). If published, this will include your full peer review and any attached files.

Reviewer #1: No

Reviewer #2: No

Reviewer #3: No

---

## [Author Response · Author response to Decision Letter 0]

17 Sep 2024

Dr. Luca Bondioli, PhD

Academic Editor

PLOS ONE

Dear Dr. Bondioli,

We are very thankful for your and the reviewers’ insightful and constructive comments regarding our manuscript entitled “‘But no living man am I’: Bioarchaeological evaluation of the first-known female burial with weapons from the 10th-century-CE Carpathian Basin” that was submitted to PLOS ONE (manuscript ID: PONE-D-24-19948). The main text has been modified following the suggestions, and an additional figure (Figure 12) has been created and added. Additionally, new references have been included, and the numbering has been adjusted accordingly. Please note that since these new references were created using Mendeley Desktop, Microsoft Word did not systematically track the updates in the Reference section or their corresponding citations in the main text. Therefore, we have highlighted these new additions with a yellow background color in the tracked version of the manuscript. The revised files have been uploaded to the submission site of the journal. We are sure that you and the reviewers helped us to improve the quality of our manuscript.

Responses to the comments and questions of Reviewer #1

Reviewer: The research paper, entitled "But no living man am I": Bioarchaeological evaluation of the first-known female burial with weapons from the 10th-century CE Carpathian Basin" by Balázs Tihanyi and co-authors presents the bioarchaeological study of a female burial (No 63), discovered within the 10th century CE century CE cemetery of Sárrétudvari-Hízóföld (eastern Hungary). The tomb yielded a grave goods set that is related to both the female sphere (including ornaments that recurred in other female and subadult tombs at the site) and the male sphere (including weapons and an archery set, which were typical of the Sárrétudvari-Hízóföld male tombs). The study of this funerary context addresses a complex and contentious topic: the identification of female warrior status in past human societies through the examination of funerary evidence. Furthermore, it considers the question of determining an individual's sex and gender based on biological parameters and material culture. The authors are fully aware of the critical nature of the topic and I was pleased to see a lengthy paragraph in the introduction to the paper outlining the state of the art of approaches to the study of gender identity and sex determination from an archaeological and bioarchaeological perspective. In general, this article is well written, the English used is fluid and the approach to the scientific data is critical. In the presentation of the 'palaeopathological' data and subsequent discussions, the authors adopt a nuanced approach, acknowledging the limitations of the sample and the multi-etiology of the lesions under consideration.

Response: We are grateful for the Reviewer’s insightful comments and suggestions, which have provided invaluable guidance in improving our manuscript. We particularly appreciate your recognition of our approach to the critical topic of sex and gender determination in archaeological contexts, as well as our careful consideration of the limitations inherent in paleopathological analysis. Below, we address each of the Reviewer’s notes in detail.

Reviewer: However, I would recommend that the authors consider more explicitly that, with the exception of the fractures at the humeroscapular joint, the observed lesions may be more likely to be related to the mature age of the individual and not to a specific activity associated with.

Response: We agree with the Reviewer that the observed changes are not specific to a particular activity, and their interpretation is indeed limited by the multifactorial etiology, poor bone preservation, and lack of comparative data on other females, with or without weapon in their grave. These limitations are highlighted in the Discussion and conclusions section, in lines 602–621 and 633–650 of the original text. That being said, physical stress remains considered one of the main factors influencing the development of entheseal changes (e.g., Karakostis et al. 2019).

In the case of SH-63, non-mechanical factors such as the advanced age of the individual alone cannot fully explain the development of these changes, particularly given the documented asymmetry in some of the extant entheseal sites and the presence of various changes (as illustrated in Figure 10 of the original text). For instance, the groove at the enthesis of the latissimus dorsi/teres major muscles observed only on the right humerus of SH-63 (illustrated in Figure 10 C of the original text) is characteristic formation more commonly seen in sub-adult individuals, though it also appears in adults. In recent years, a research group led by Olivier Dutour and Hélène Coqueugniot has begun analyzing groove-like formations of the latissimus dorsi/teres major muscle enthesis in sub-adults and young adults using microCT. Their results on the microstructure of these entheses (e.g., Navarre et al. 2023; Navarre et al. 2024), ruled out developmental causes and proposed two hypotheses related to physical activities: a) variations in the adaptive response of the bone in connection with the growth stage, and b) variations in the nature and intensity of biomechanical constraints in connection with age. These findings are promising and offer valuable insights for evaluating entheseal changes. 

However, a cautious interpretation remains necessary in the case of SH–63. This is why we used phrases like ‘could have’ or ‘may have had’ in the Discussion and conclusions section when referring the possible lifestyle of SH-63. Despite the recent findings on entheseal changes and existing similarities with the pattern observed in other individuals with weapon equipment, we emphasize that further research is required to fully understand these phenomena and confirm any lifestyle-related implications.

References:

Karakostis, F. A., Jeffery, N., & Harvati, K. (2019). Experimental proof that multivariate patterns among muscle attachments (entheses) can reflect repetitive muscle use. Scientific Reports, 9, 16577. https://doi.org/10. 1038/s41598-019-53021-8

Navarre, G., Dutailly, B., Vanderesse, N., Dutour, O., & Coqueugniot, H. (2024). Effet de l’âge sur l’organisation de l’os trabéculaire en cas d’ostéolyse corticale d’insertion (OCI) Analyse 3D-μCT des changements enthéséaux sur un échantillon ostéoarchéologique juvénile. Groupe Des Paleopathologistes de Langue Française. Colloque 2024.

Navarre, G., Dutour, O., & Coqueugniot, H. (2023). Les enthésopathies en creux: développement ou surcharge mécanique? Groupe Des Paleopathologistes de Langue Française. Colloque 2023, 20.

Reviewer: Moreover, the presence of the archer's set in the tomb (an element that, in a post-processualist perspective, could also be indicative of a 'gift' or the exaltation of a status and not of the deposition of the individual personal objects) is worthy of further consideration.

Response: We agree with the Reviewer concerning the complex symbolism of artifacts in burials. We also acknowledge and align with the methodological considerations of scholars such as Heiko Steuer, Sebastian Brather, and Heinrich Härke, who argue that the evaluation of artifacts does not necessarily reflect the status and lifestyle of the deceased directly. This perspective is highlighted in the Introduction (lines 106–116) and the Discussion and conclusions (lines 742–758) sections of the original text.

In Hungarian archaeology, the relationship between the deceased, the burial customs applied, and the grave goods provided indeed requires further research. However, the most recent overview of the 10th-century-CE burials in the Carpathian Basin (referred to as Conquest Period burials) pointed out that artifacts – including clothing elements, jewelry, weapons, and prestigious items – that were specifically crafted and provided for burial and funeral purposes are exceptionally rare in Conquest Period burials (Révész 2020). Furthermore, the investigation of Conquest Period burials primarily focuses on the composition and characteristics of the grave goods and related burial customs, as burial constructions are generally absent (with the exception of traces of wooden coffins), and grave pits do not exhibit significant variation in their size and forms (Révész 2020).

In our case, even if the provided archery equipment did not belong to the personal possessions of SH-63, it does not alter the core findings of our study – namely, that a weapon was found in a 10th-century-CE female grave from the Carpathian Basin, with burial customs similar to those of its male counterparts. This is a significant factor when evaluating the possible social/military status or ‘occupation’ of the individual. However, as stated in the Discussion and conclusions section (lines 742–758 of the original text), this remains an open debate due to methodological problems and the lack of comparative data.

To incorporate your valuable feedback, we have revised and nuanced the Discussion and conclusions sections and we have highlighted the methodological challenges concerning the symbolic nature of grave goods.

Original text (lines 717–720 of the original text):

“Additionally, the location of the bow remain in the grave suggests a connection with the deceased, as the plate was in a rather functional position, as if ‘she was holding it in her left hand’, such as described by the archaeologist in charge of the excavation of the cemetery [148].”

Revised text (lines 705–709):

“While the origin and symbolic significance of these artifacts (e.g., personal items or gifts) remain unclear, the location of the bow remain in the grave suggests a connection with the deceased, as the plate was in a rather functional position, as if ‘she was holding it in her left hand’, such as described by the archaeologist in charge of the excavation of the cemetery [149].”

Original text (lines 750–758 of the original text):

“Certainly, one of the most intriguing questions is whether the case can be considered a warrior burial. Unfortunately, at the current research level, this must remain an open debate. First, the term “warrior” involves specific aspects on social and legal levels for which sources other than archaeological and anthropological are also required for identification, but no written data are available concerning warrior women among the Magyars in the 10th century CE. Furthermore, in nomadic tribes of the eastern steppes, similar to the early Magyars, it was common for females to learn how to defend themselves and the livestock to survive [141,142]. While this probably resulted in practicing similar daily activities with males, they were not necessarily considered as dedicated warriors.”

Revised text (lines 739–749):

“Certainly, one of the most intriguing questions is whether the case can be considered a warrior burial. Unfortunately, at the current research level, this must remain an open debate. First, the term “warrior” involves specific aspects on social and legal levels for which sources other than archaeological and anthropological are also required for identification, but no written data are available concerning warrior women among the Magyars in the 10th century CE. Additionally, the origin and symbolic significance of the artifacts — whether they were the personal items of the deceased or gifts from the community — remain unclear. Furthermore, in nomadic tribes of the eastern steppes, similar to the early Magyars, it was common for females to learn how to defend themselves and the livestock to survive [142,143]. While this probably resulted in practicing similar daily activities with males, they were not necessarily considered as dedicated warriors”

Reference:

Révész, L. (2020). A 10–11. századi temetők regionális jellemzői a Keleti-Kárpátoktól a Dunáig. Szegedi Tudományegyetem Régészeti Tanszéke–Magyar Tudományos Akadémia Régészeti Intézete–Magyar Nemzeti Múzeum–Martin Opitz Kiadó.

Reviewer: Additionally, as proposed below, I would like the authors to extend the comparison between the bioarchaeological evidence gathered in this study on individual SH 63 and that available not only on the other individuals with weapons in the tombs, but also on the other female individuals and, in general, for the necropolis as a whole. This would help to avoid forcing a connection (tombs with weapons) that could produce flawed observations.

Response: We appreciate the Reviewer's suggestion to extend the comparison. We agree that avoiding a forced connection between tombs with weapons is essential to prevent potentially flawed conclusions.

The primary limitation of our interpretation lies in the uniqueness of SH-63 and the current lack of comparative data, as highlighted in lines 759–760 of the original text. Although data on pathological and supposed activity-related changes in males with and without weapons from the Sárrétudvari–Hízóföld cemetery has increased over the last decade (e.g., Berthon et al. 2015; Tihanyi et al. 2015; Berthon 2019; Berthon et al. 2019; Tihanyi et al. 2020; Tihanyi 2020), similar data on female individuals remains highly limited. The absence of data on females highlights the need for extensive future research to build a comprehensive database, including data on female individuals from various 10th-century-CE series of the Carpathian Basin. At present, comparison is only possible with data published in the 1990s (e.g., Pálfi 1992; 1993; Pálfi & Dutour 1996; Pálfi 1997), and we must acknowledge that methodologies, particularly in the diagnosis of various pathological changes, have evolved over the intervening decades.

In the Sárrétudvari–Hízóföld series, six cases with possible traces of osteoporosis were described, five of which were identified as adult females, while one individual was classified as indeterminate adult (Pálfi 1993; 1997). In contrast, no osteoporosis-related changes were documented in adult males. In addition, traumatic lesions, which were frequently observed in the series, were typically associated with males, with only a few cases registered in females. This suggests differences in lifestyle between the sexes (e.g., Pálfi 1992; 1993; Pálfi & Dutour 1996). Similarly, entheseal and joint changes were predominantly found in males (e.g., Pálfi 1993; Pálfi & Dutour 1996; Pálfi 1997), with entheseal changes of the upper limb bones reported in 21 males but only in one female (Pálfi 1997).

Following the Reviewer’s suggestion, we have extended the relevant parts of the Discussion and conclusions section. However, we have done so cautiously, addressing the above-described methodological issues and focusing on the trends observed by earlier scholars in the Sárrétudvari–Hízóföld series.

Changes applied to the Discussion and conclusions section:

1. Osteoporosis (including changes requested by Reviewer #2):

Original text (lines 547–555 of the original text):

"Several antemortem features are present on the skeletal remains of SH-63. Firstly, we observed that the bones are much lighter than in the case of other skeletons from the Sárrétudvari–Hízóföld series, possibly due to osteopenia, a loss of bone mass. Particularly, bone fragility, traces of a reduced trabecular system in the vertebrae, an increased diameter of the medullary cavity in the long bones, thinning of cortical bone in both the skull and postcranial elements, and antemortem bone fractures suggest the presence of osteoporosis in the skeleton (e.g., [165,179–182]). The presence of osteoporosis, described as a disease that typically affects older women [165,180], indirectly supports the results concerning the sex determination of SH-63.”

Revised text (lines 500–521):

“Several antemortem features are present on the skeletal remains of SH-63. Firstly, we observed the lightweight nature of the bones, which could indicate osteopenia, a loss of bone mass. Specifically, bone fragility, traces of a reduced trabecular system in the vertebrae, an increased diameter of the medullary cavity in the long bones, thinning of cortical bone in both the skull and postcranial elements, and antemortem bone fractures may be associated with the presence of osteoporosis in the skeleton (e.g., [166,174–177]). The presence of osteoporosis, described as a

---

## [Decision Letter · Decision Letter 1]

4 Nov 2024

‘But no living man am I’: Bioarchaeological evaluation of the first-known female burial with weapon from the 10th-century-CE Carpathian Basin

PONE-D-24-19948R1

Dear Dr. Tihanyi,

We’re pleased to inform you that your manuscript has been judged scientifically suitable for publication and will be formally accepted for publication once it meets all outstanding technical requirements.

Kind regards,

Luca Bondioli, PH.D.

Academic Editor

PLOS ONE

Additional Editor Comments (optional):

Reviewers' comments:

Reviewer's Responses to Questions

**Comments to the Author**

1. If the authors have adequately addressed your comments raised in a previous round of review and you feel that this manuscript is now acceptable for publication, you may indicate that here to bypass the “Comments to the Author” section, enter your conflict of interest statement in the “Confidential to Editor” section, and submit your "Accept" recommendation.

Reviewer #2: All comments have been addressed

Reviewer #3: All comments have been addressed

2. Is the manuscript technically sound, and do the data support the conclusions?

Reviewer #2: Yes

Reviewer #3: Yes

3. Has the statistical analysis been performed appropriately and rigorously? 

Reviewer #2: N/A

Reviewer #3: N/A

4. Have the authors made all data underlying the findings in their manuscript fully available?

Reviewer #2: Yes

Reviewer #3: Yes

5. Is the manuscript presented in an intelligible fashion and written in standard English?

Reviewer #2: Yes

Reviewer #3: Yes

6. Review Comments to the Author

Reviewer #2: Dear authors,

Thanks for addressing my comments - I think the changes made have improved the quality and credibility of case made. Regarding the requests for edits that the authors decided not to make, I think the authors explained their rationale well. Although I still hold the opinion that some of the suggested edits would have benefitted the manuscript (e.g. suggested changes to the title, and the section on osteoporosis and described lesions), I do see the side of the authors for their choice, and it not an issue to me. I do not have further comments to add.

Best of luck with the finalization of the manuscript and the continuation of the project.

Reviewer #3: The authors have addressed all of my comments, as well as those of the other reviewers (as far as I can judge), thus I recommend the paper for publication.

7. PLOS authors have the option to publish the peer review history of their article (what does this mean?). If published, this will include your full peer review and any attached files.

Reviewer #2: **Yes: **Simone Anna Maria Lemmers

Reviewer #3: No

---

## [Editor Report · Acceptance letter]

15 Nov 2024

PONE-D-24-19948R1 

PLOS ONE

Dear Dr. Tihanyi, 

I'm pleased to inform you that your manuscript has been deemed suitable for publication in PLOS ONE. Congratulations! Your manuscript is now being handed over to our production team.

Kind regards, 

on behalf of

Dr. Luca Bondioli 

Academic Editor

PLOS ONE